# Fiber Optic Sensor with a Gold Nanowire Group Array for Broad Range and Low Refractive Index Detection

Gongli Xiao [1], Jiapeng Su [1], Hongyan Yang [2,*], Zetao Ou [1], Haiou Li [1], Xingpeng Liu [1], Zanhui Chen [1], Yunhan Luo [3] and Jianqing Li [4]

[1] Guangxi Key Laboratory of Precision Navigation Technology and Application, Guilin University of Electronic Technology, Guilin 541004, China
[2] Guangxi Key Laboratory of Optoelectronic Information Processing, School of Optoelectronic Engineering, Guilin University of Electronic Technology, Guilin 541004, China
[3] College of Science and Engineering, Jinan University, Guangzhou 510632, China
[4] Guangdong-Hong Kong-Macao Joint Laboratory for Intelligent Micro-Nano Optoelectronic Technology, Macau University of Science and Technology, Macau 999078, China
* Correspondence: hyyang@guet.edu.cn; Tel.: +86-177-7732-7521

**Abstract:** To achieve high performance and wide range detection, we propose an ultra-wide range high sensitivity plasmonic fiber optic sensor with a gold (Au) nanowire group array, which has both propagating surface plasmon resonance (PSPR) and local surface plasmon resonance (LSPR) sensing characteristics. The PSPR, LSPR, and PSPR+LSPR are presented as Au thin layers, Au spheres (or Au nanowires), and Au nanowire group arrays, respectively, and their respective properties are analyzed from theoretical, simulated, and numerical aspects. When detection is performed, the presence of both evanescent wave and electric field forces in the Au nanowire group array combines to significantly improve the sensor's detection capability. Detection simulation analysis was performed using COMSOL Multiphysics software. The range of refractive indices that can be detected is 1.08 to 1.37 in the optical band from 1210 nm to 2140 nm. In the detection range, the maximum sensitivity of the detected wavelength is 13,000 nm/RIU. Our proposed sensor has a broad range, high sensitivity, and low refractive index detection, and has good research value and application prospects.

**Keywords:** Au nanowire groups array; localized surface plasmon resonance; propagating surface plasmon resonance; broad range detection; low refractive index detection; photonic crystal fiber

## 1. Introduction

The detection of low refractive index (RI) ranges is more difficult than conventional detection, and the detected range of lower RI is also smaller [1,2]. In recent years, detection using photonic crystal fibers (PCFs) has been well-regarded by researchers as the process technology has been refined, and this has allowed low RI analytes such as liquid carbon dioxide [3] and fluorine-containing organics [4] to be detected very well [5]. PCF is usually made of fused silica, polymers, or plastics as the background material and has a periodic or nonperiodic array of microtia extending along the full length of the fiber [6]. PCF has been applied in many fields, such as fiber optic communication [7], biological detection [8], medical diagnosis [9], and optoelectronic sensing [10]. Generally speaking, pure water has an RI of 1.33, and analytes with an RI below this value are referred to as low RI analytes [11]. Wide range detection means that the effective detection range of the sensor is large, generally referring to an RI range greater than 0.10 [12,13].

Of course, it is not enough to rely on PCF detection alone. There are more fiber optic sensors based on PSPR or LSPR [14,15]. PSPR is a quasiparticle description of the collective resonant motion of electrons on metal surfaces, occurring at the dielectric–metal interface. In PSPR-PCF, metal is generally applied to the sensing area in the form of a thin layer [16]. Hasan Abdullah et al. proposed a highly sensitive Au-coated circular

nanofilm PCF biosensor based on SPR [17]. The design enhances the SPR by applying the Au layer to the periphery of the optical fiber to achieve high-performance sensor detection. A surface plasmon resonance (SPR) sensor based on D-type photonic crystal fiber was proposed by Han Liang et al. [18]. The performance of SPR-PCF sensors coated with a graphene layer and a zinc oxide layer was evaluated using silver as a plasmonic metal. The LSPR phenomenon describes the propagation of surface plasmons excited in sizes smaller than nanoparticles or nanograting [19]. In LSPR-PCF, metal is generally applied to the sensing area in the form of small balls or metal wires. Shengxi Jiao et al. proposed an LSPR-PCF sensor with silver nanowires deposited in microchannels, and the detection advantages of LSPR were well illustrated by comparing the silver nanowires with the same standard silver thin layer [20]. D. Paul and R. Biswas proposed an LSPR-PCF sensor based on Au nanospheres to enhance the detection sensitivity and range [21]. Through theory and simulation, it is demonstrated that Au nanospheres can effectively enhance sensor detection performance. As the research progresses, the disadvantages of PSPR and LSPR are also revealed. PSPR has an inherent ohmic loss, which limits the propagation of evanescent waves, and the concentration of electrons is smaller because of the large cross-sectional area of the thin metal layer [22,23]. A single LSPR does not produce significant evanescent waves, and has limited detection capability [24]. Therefore, to address the above limitations, we effectively combine PSPR and LSPR with both effects to reduce the limitations and enhance the sensor detection performance.

In recent years, with the improvement in the manufacturing process, more and more complex structures have been applied in the field of sensing and detection, among which the proposed metal nanowire array provides new research ideas for optical fiber sensors [25–27]. The metal nanowire array actually belongs to LSPR, but it is different from it in that the metal nanowire array is composed of multiple metal wires, which may have new detection properties along with the traditional LSPR properties. Pathak, A.K. et al. proposed a highly sensitive bio-detection by embedding metal nanowires into a dual-hole microchannel. The design utilizes two Au nanowires to excite two modes of coupling to achieve high performance detection characteristics with a sensitivity of 90,500 nm/RIU at an RI of 1.40 [28]. Meshginqalam, B. and Barvestani, J. proposed a highly sensitive fiber optic sensor for cancer cell detection using two sets of bimetallic wires. The design placed two sets of bimetallic wires at the upper and lower ends of the optical fiber and obtained good detection performance after rational optimization of the structural parameters, and this sensor can be used to detect six different cancer cells. The detection sensitivity for MCF-7 was 53,571 nm/RIU [29]. Zhan, Y. et al. designed a fiber optic sensor based on metal nanowire surrounds, which replaced the traditional metal thin layer with a ring-shaped metal nanowire, which improved the detection performance of the sensor. It has a detection range of 1.33–1.40 and a maximum sensitivity of 12,314 nm/RIU [30]. These metal nanowire structures offer a greater improvement in detection performance than conventional thin-layer sensors. However, they also have some drawbacks, such as how to fix the nanowires smoothly and how to enhance the detection range of this type of sensor. In conclusion, metal nanowire arrays have good research value in the field of detection.

In this paper, we propose an Au nanowire group array as a sensing region for fiber optic sensors. The sensor is D-type and the detection plane makes it easier to fix gold nanowires. This structure has the characteristics of PSPR and LSPR, and is compared with the conventional PSPR and LSPR structures in terms of mechanism, simulation, and numerical analysis, which fully demonstrates the good detection ability of the proposed Au nanowire groups array in the low RI range, and provides new ideas for other research directions.

## 2. Model and Principle Analysis

### 2.1. Model Analysis

Figure 1 shows the fiber optic sensor structure model. The sensing plane is polished above the distance $d_h = 1\ \mu m$ from the center of the fiber, a layer of $TiO_2$ with thickness $h_1 = 10\ nm$ is first applied to the sensing plane, and then array of Au nanowire groups is placed on it, each of which is composed of five identical Au nanowires in close proximity. Each nanowire has same diameter of $d = 70\ nm$, and every fifth Au nanowire forms the Au nanowire group. In the sensing plane, there are 12 groups of Au nanowires forming an array of Au nanowire groups, with only the outermost and second outermost Au nanowire groups on both sides have a spacing of $\Lambda_2 = 2d = 140\ nm$, and the other Au nanowire groups have a spacing of $\Lambda_1 = d = 70\ nm$. The stomata in the fibers are arranged in a stacked manner, with the diameter of each air-hole being $d_a = 2\ \mu m$ and the spacing of each air-hole being $\Lambda_a = 4.5\ \mu m$. The outermost layer of the fiber is an ideal cladding, which we can consider as a perfect match layer (PML).

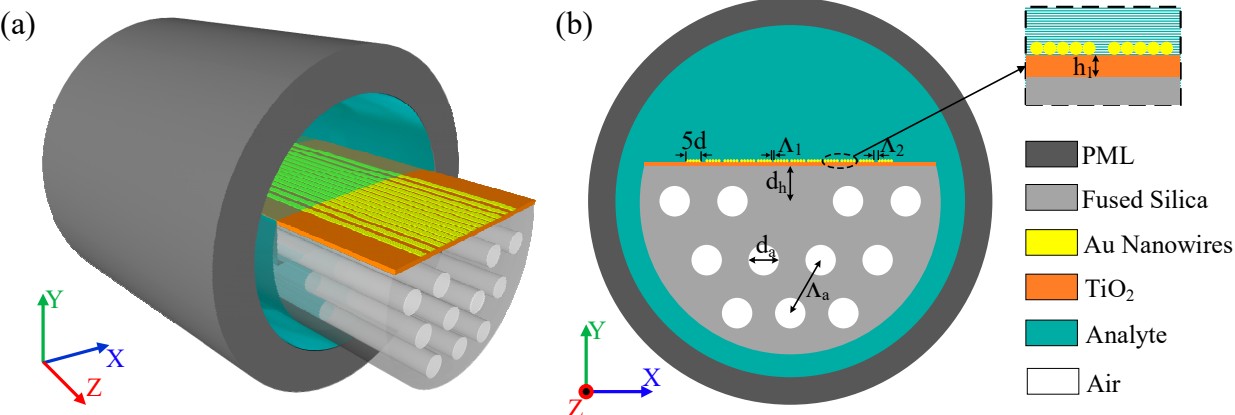

**Figure 1.** Fiber optic sensor structure model. (**a**) Three-dimensional model; (**b**) two-dimensional cross section.

This article covers a variety of materials, and we use the RI or dielectric constant of the material to express the properties of each material. The RI of silica can be introduced using the Sellmeier formula [31], the dielectric constants of Au can be expressed in the Drude–Lorenz model [32,33], and the RI of $TiO_2$ can be expressed as follows [34]:

$$n_{TiO_2}^2 = 5.913 + \frac{2.441 \times 10^7}{(\lambda^2 - 0.803 \times 10^7)} \tag{1}$$

where $n_{TiO_2}$ is the RI of TiO$_2$ and $\lambda$ is the wavelength of the incident light. The RI of air is normally 1 [32].

The constraint loss is a very important part of the analysis of sensing performance, and its equation is expressed as follows [34]:

$$\alpha_{Loss}(\lambda) = \frac{40\pi}{\lambda \ln 10} Im\left(n_{eff}\right)\ [dB/cm] \tag{2}$$

where $Im\left(n_{eff}\right)$ is the imaginary part of the effective RI. The level of sensitivity is an important indicator of sensor performance. Wavelength sensitivity (WS) is usually used to reflect the performance of the sensor. The expression of WS is as follows [35]:

$$S_\lambda = \frac{\Delta\lambda_{peak}}{\Delta n_s}\ [nm/RIU] \tag{3}$$

where $\Delta\lambda_{peak}$ is the offset of the resonance wavelength (RW) between adjacent RI and $\Delta n_s$ is the amount of change in the RI of the analyte.

### 2.2. Principle Analysis

Analyzing things mechanistically is very important. Although both PSPR and LSPR are based on SPR, there are many differences between them. Figure 2 shows the detection principles of PSPR, LSPR, and the combined Au nanowire group array of both. In Figure 2a, PSPR is a kind of evanescent wave propagating along the metal–dielectric interface. The electrons on the surface of the metal film move collectively and resonantly at the metal–dielectric interface, coupling with the incident light and propagating continuously forward along the partition interface. Therefore, some of the electrons in PSPR are uniformly distributed at the metal–dielectric interface and the rest are distributed at the ends of the thin layer under the action of evanescent wave. It also has shortcomings; because the area of the thin layer is relatively large, it is difficult for the evanescent wave to cover the entire thin layer, and the inherent ohmic loss problem cannot be avoided. Figure 2b shows the LSPR, where the incident light deflects the electrons within the surface of the region, while the Coulomb force of the nucleus on the electrons also deflects them, and the electrons move back and forth in the equilibrium position under the combined effect of the two forces. Therefore, the electrons within the localized surface in the LSPR are distributed at the top and bottom of the structure. However, the interaction between the individual metal nanospheres is very small, limiting the detection performance of LSPR.

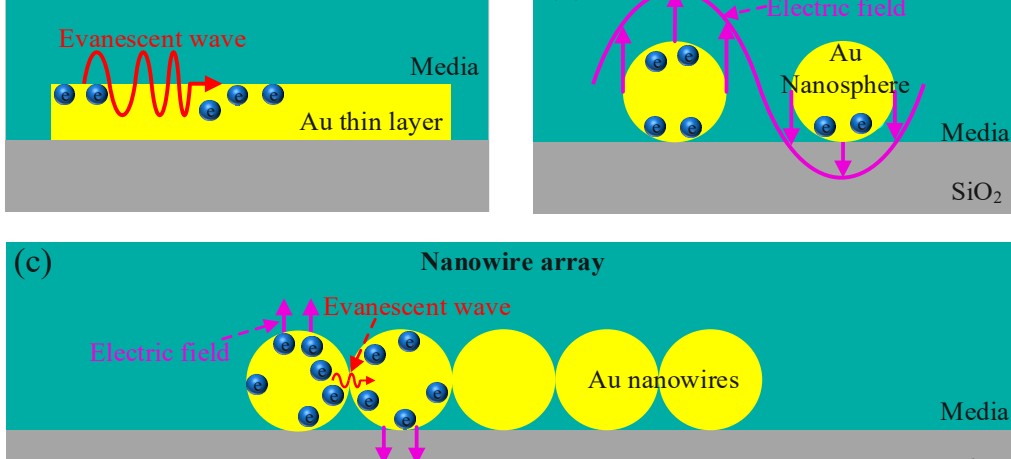

**Figure 2.** (**a**) PSPR sensing principle; (**b**) LSPR sensing principle; (**c**) Au nanowire group array sensing principle.

Figure 2c shows the sensing mechanism of the Au nanowire group array. The Au nanowire arrays are composed of multiple Au nanowires, which contain two properties. The first property is similar to PSPR, because multiple Au nanowires are placed close to each other, so the part of two Au nanowires touching each other can be regarded as a very thin layer of Au film, so that the electrons in different Au nanowires can be influenced by the evanescent wave, and thus move. Because the contact point is small, some electrons will gather at the connection of the two Au nanowires. Another property similar to LSPR, each Au nanowire group is also subjected to electromagnetic fields that cause electrons to cluster at the upper and lower ends of the Au nanowires. Our study of PSPR+LSPR sensing properties has been described in other related literature, where continuous and contacting Au nanowires can be used to generate PSPR and the upper and lower ends of each Au nanowire can generate LSPR, the combination of which achieves an effective enhancement of SPR [22,23]. In the following, we also analyze the relevant properties from simulations. Therefore, from the above mechanism analysis, the Au nanowire group array has high detection characteristics.

### 3. Simulation Discussions and Results

To ensure the sensor design is reasonable and the calculation is accurate, we analyze the model based on the finite element method (FEM) and use COMSOL Multiphysics software simulation and calculations.

### *3.1. Mode Analysis and Dispersion Relations*

Light propagation in optical fibers can be solved by the fluctuation equation, and the longitudinal and transverse field solutions in optical fibers are different, so this requires a mode analysis of the fiber. Figure 3 shows the mode analysis and dispersion relationship of the sensor. The RI of the analyte is 1.34 for the sensor, and all parameters are the same as the model in Figure 1. When most of the light propagates in the center of the fiber, the polished detection plane has no light intensity, and this mode is called the core mode. Conversely, when there is almost no light in the core and most of the energy is gathered in the detection plane, this mode is called surface plasmon polaritons (SPPs) mode. The real part of the effective refraction of the two modes varies with the wavelength of the incident light. When the incident light is a certain value, the two modes will be phase matched to produce a strong coupling, and the real part of the two modes, which was originally monotonically decreasing, produces a small kink near this wavelength. At the same time, a large amount of energy in the core flows to the detection plane. The wavelength that can produce these phenomena is called the resonance wavelength. I, II, and III in Figure 3 show the core mode, SPP mode, and phase-matched mode, respectively. Currently, the fiber core has more energy and the detection surface also has energy. Different analytes have different refractive indices, the wavelengths of the resonance are different, and the detection of analytes is achieved by analyzing different resonance wavelengths.

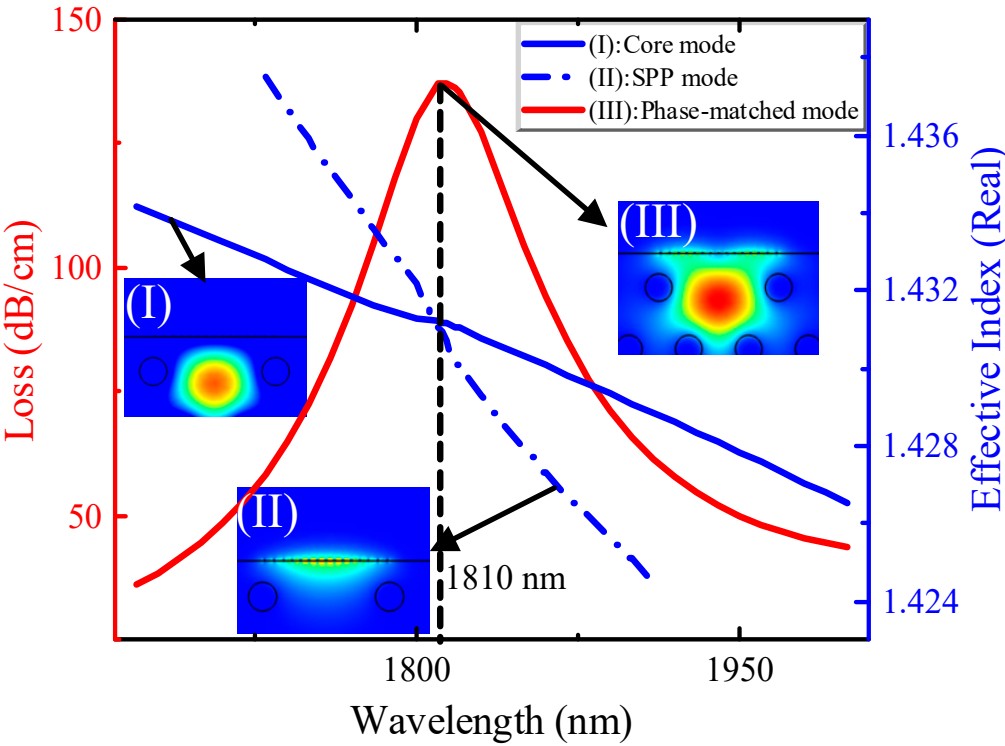

**Figure 3.** Dispersion relationship and mode analysis of the sensor.

### *3.2. Simulation Analysis of Sensing Area*

We analyzed the properties of PSPR, LSPR, and Au nanowire group array mechanically, and to better prove them, we simulated their electric fields and the carrier distribution in the *X*-axis direction by software simulation. As shown in Figure 4a, (I) and (II) are the middle part of the Au thin layer and one side of the endpoint; in the electric field diagram,

it can be seen that the electric field in the middle part of the Au thin layer is more uniformly distributed, because the Au thin layer is longer, so the electric field at both ends is smaller. From the *X*-axis direction, the carrier concentration can be seen, as well as electrons in the metal Au thin layer uniform distribution. In Figure 4b, (III) and (IV), the middle part of the Au nanowire group and the endpoint are shown on one side. It can be seen from the electric field diagram that the electric field is stronger at the top and bottom of the middle part of the Au nanowire group, and a larger electric field also exists on the Au nanowire on its side. From the carrier concentration in the *X*-axis direction, there are more electrons at the connection of each Au nanowire. In Figure 4c, (V) and (VI) are the middle part of the Au nanowire group and the endpoint on one side. From the electric field diagram, there is a stronger electric field at the ends of the Au nanowire group and the contact region of the Au nanowires has a stronger electric field, as seen from the carrier concentration in the *X*-axis direction, where more electrons are gathered at the junction of each Au nanowire. Observing Figure 4b,c, we can find that there is a large electric field and displacement current density at the contact points of the adjacent Au nanowires. This is because, not only is there evanescent wave controlling the charge movement at the contact point, but also near the contact point, owing to the excessive high energy and very small distance, there may be a quantum tunneling effect between the two Au nanowires, which in turn increases the charge flow between the Au nanowires [36,37].

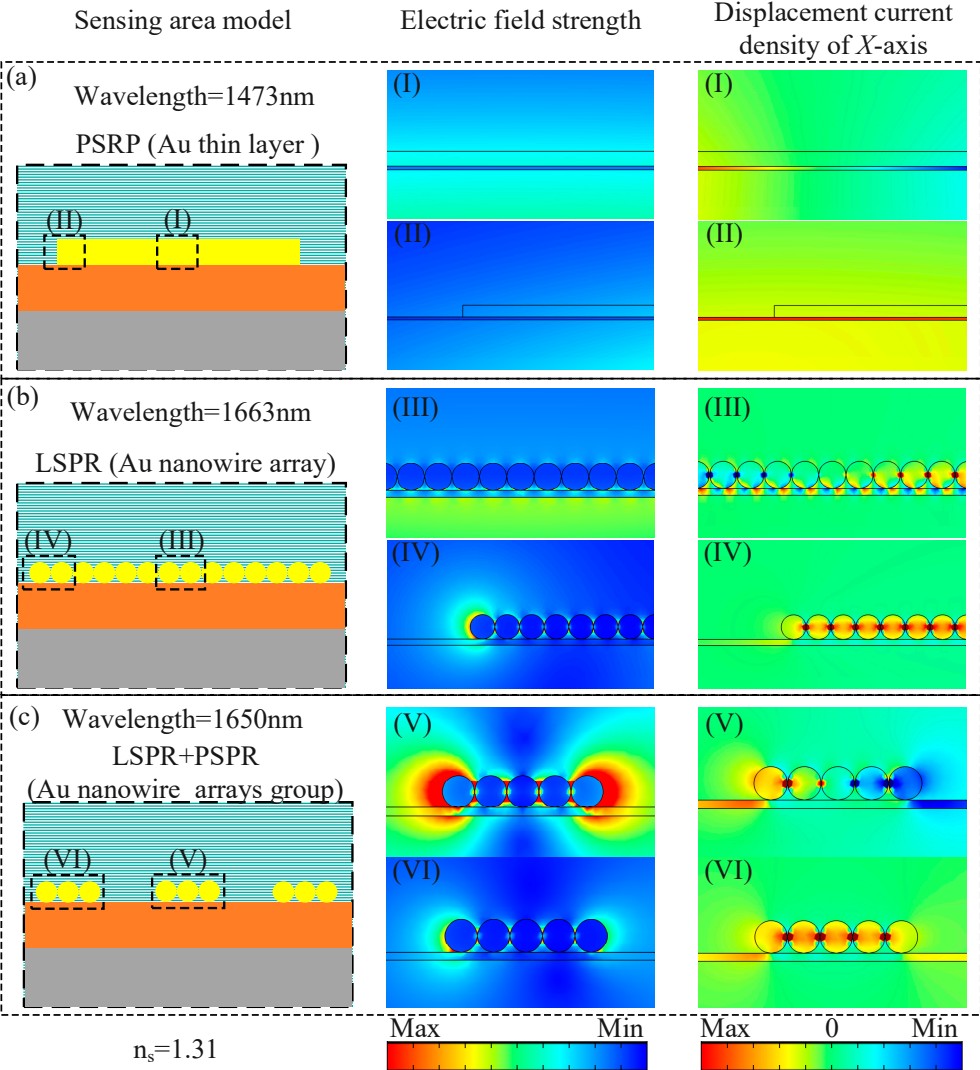

**Figure 4.** Simulation analysis of sensing area. (**a**) PSPR sensing principle; (**b**) LSPR sensing principle; (**c**) Au nanowire groups array sensing principle.

We illustrate the advantages of the Au nanowire group array through mechanism and simulation analyses. Numerical analysis is also an essential part. Figure 5a–d show the numerical analysis of the induction zone with different structures. When the refractive indices are 1.30 and 1.31, we simulated the Au thin layer, the Au nanowire array, the equally spaced Au nanowire groups array, and the adjusted spacing Au nanowire group array, respectively. Under the same conditions, they were detected with a WS of 2200 nm/RIU, 2800 nm/RIU, 3700 nm/RIU, and 4000 nm/RIU, respectively. Numerically, we found that the spacing-adjusted Au nanowire arrays were about 81% more sensitive than the conventional Au thin layer in terms of the detection region. In addition, compared with the Au nanowire array with the same spacing, the spacing adjustment can make the Au nanowire array act more effectively at the right location and resonate with more incident light, further enhancing the detection capability of the sensor.

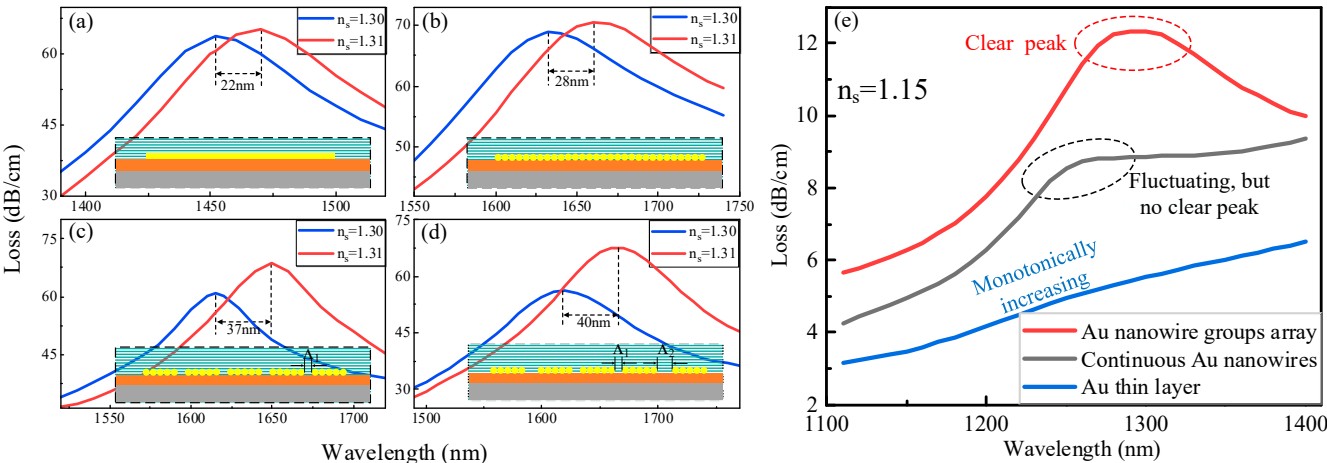

**Figure 5.** Loss analysis for different sensing regions at analytes 1.30 and 1.31. (**a**) Au thin layer; (**b**) Au nanowires; (**c**) Au nanowire groups array with $\Lambda_1 = 70$ nm spacing; (**d**) Au nanowire group array with $\Lambda_1 = 70$ nm and $\Lambda_2 = 140$ nm spacing. Detection range analysis (**e**) Au thin layer, continuous Au nanowires, and Au nanowire group array.

In addition to the improvement in sensitivity, the Au nanowire group array can also achieve an improvement in the detection range. As shown in Figure 5e, when the RI is 1.15, the loss spectra of Au thin layer, continuous Au nanowires, and Au nanowire group arrays are analyzed under the premise of ensuring the same substitution parameters. When the RI is low, the detection ability of the Au thin layer is limited, and its loss curve is monotonically increasing, and the change is not sufficiently obvious to achieve detection in this RI. In the continuous Au nanowire structure in this RI, there is a significant fluctuation, but it did not form a clear wave peak, and thus detection also cannot be well achieved. The Au nanowire group array structure exhibits a clearer wave peak at this RI, and although the data are not very good, there are effective peaks at low refractive indices. Therefore, the Au nanowire group array designed by the authors can appropriately increase the detection range of the sensor.

Through the above simulation and numerical comparison analysis, it is more effectively proven that the Au nanowire group arrays effectively integrate the characteristics of PSPR and LSPR, further enhancing the detection range and detection capability, and laying a solid theoretical and simulation foundation for achieving wide range and high sensitivity detection.

### 3.3. Structural Parameter Optimization

The numerical optimization of the structure can further improve the detection capability of the sensor. In our proposed sensor model, the diameter of the Au nanowires has a large effect on the sensitivity, so we have performed a parametric optimization analysis by taking the diameter. To ensure a clearer observation of the relationship between Au

nanowire diameter and sensitivity, and to ensure a more suitable analyte for the subsequent experimental phase, we chose an analyte with a RI of 1.33 to 1.34. In Figure 6, the loss curves for different Au nanowire diameters at refractive indices of 1.33 and 1.34 are shown, and the other parameters are the same as those shown in Figure 1. The diameter of the Au nanowire is selected from 50 nm to 90 nm; after simulation, when the diameter is 70 nm, the RW of 1.33 RI is 1770 nm and the RW of 1.34 is 1840 nm, while the WS at this diameter is 7000 nm/RIU, which is optimal in the selected diameter range, so the diameter of Au nanowire is determined to be 70 nm.

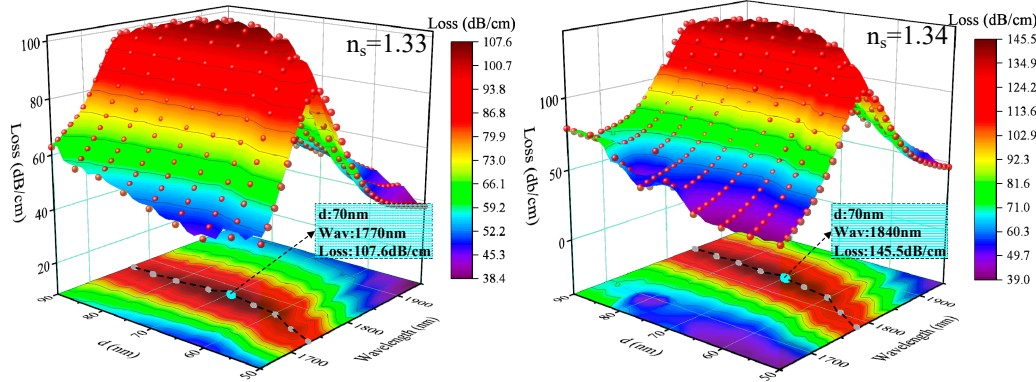

**Figure 6.** Diameter analysis of Au nanowires with RI of 1.33 and 1.34 in the analytes.

### 3.4. Sensor Performance

After optimizing the structural parameters of the sensor, we analyzed the performance of the sensor. The RI detection range of our designed sensor can reach 1.08 to 1.37. To describe the wide range of detection performance more accurately, we divided the detection bands into three: 1.08 to 1.16, 1.17 to 1.26, and 1.27 to 1.37, respectively. As shown in Figure 7, the resonance peaks as well as the shifts can be observed in the RI range from 1.08 to 1.16, although the loss curves vary less for different refractive indices, and in this range, the maximum (max) WS is 2000 nm/RIU, the average (ave) WS 1333 nm/RIU, and the fitted curve for the resonance wavelength is $y = 984x + 138$, with an adjusted $R^2$ of 99.07%. When the RI range is 1.17 to 1.26, the max WS is 4000 nm/RIU, the ave WS is 2000 nm/RIU, and the fitting curve of resonance wavelength is $y = 1667x + 636$, while the adjusted $R^2$ is 99.15%. When the RI range is 1.27 to 1.37, the max WS is 13,000 nm/RIU, the ave WS is 6300 nm/RIU, the fitted curve of resonance wavelength is $y = 43356x^2 - 108461x + 69336$, and the adjusted $R^2$ is 99.65%. Detailed data are shown in Table 1.

**Table 1.** Performance analysis of sensor.

| $n_s$ | RW (nm) | Loss (dB/cm) | WS (nm/RIU) | $n_s$ | RW (nm) | Loss (dB/cm) | WS (nm/RIU) |
|---|---|---|---|---|---|---|---|
| 1.08 | 1210 | 8.97 | 1000 | 1.23 | 1410 | 24.22 | 2000 |
| 1.09 | 1220 | 9.33 | 1000 | 1.24 | 1430 | 26.46 | 2000 |
| 1.10 | 1230 | 9.73 | 1000 | 1.25 | 1450 | 29.04 | 2000 |
| 1.11 | 1240 | 10.16 | 1000 | 1.26 | 1470 | 32.37 | 4000 |
| 1.12 | 1250 | 10.62 | 2000 | 1.27 | 1510 | 36.35 | 3000 |
| 1.13 | 1270 | 11.34 | 1000 | 1.28 | 1540 | 41.33 | 4000 |
| 1.14 | 1280 | 11.71 | 1000 | 1.29 | 1580 | 47.72 | 4000 |
| 1.15 | 1290 | 12.33 | 1000 | 1.30 | 1620 | 56.17 | 4000 |
| 1.16 | 1300 | 13.34 | 2000 | 1.31 | 1660 | 67.51 | 6000 |
| 1.17 | 1320 | 15.21 | 1000 | 1.32 | 1720 | 83.63 | 5000 |
| 1.18 | 1330 | 16.48 | 2000 | 1.33 | 1770 | 107.62 | 7000 |
| 1.19 | 1350 | 17.71 | 1000 | 1.34 | 1840 | 145.64 | 8000 |
| 1.20 | 1360 | 19.06 | 2000 | 1.35 | 1920 | 215.12 | 9000 |
| 1.21 | 1380 | 20.60 | 1000 | 1.36 | 2010 | 371.76 | 13,000 |
| 1.22 | 1390 | 22.30 | 2000 | 1.37 | 2140 | 315.63 | NA |

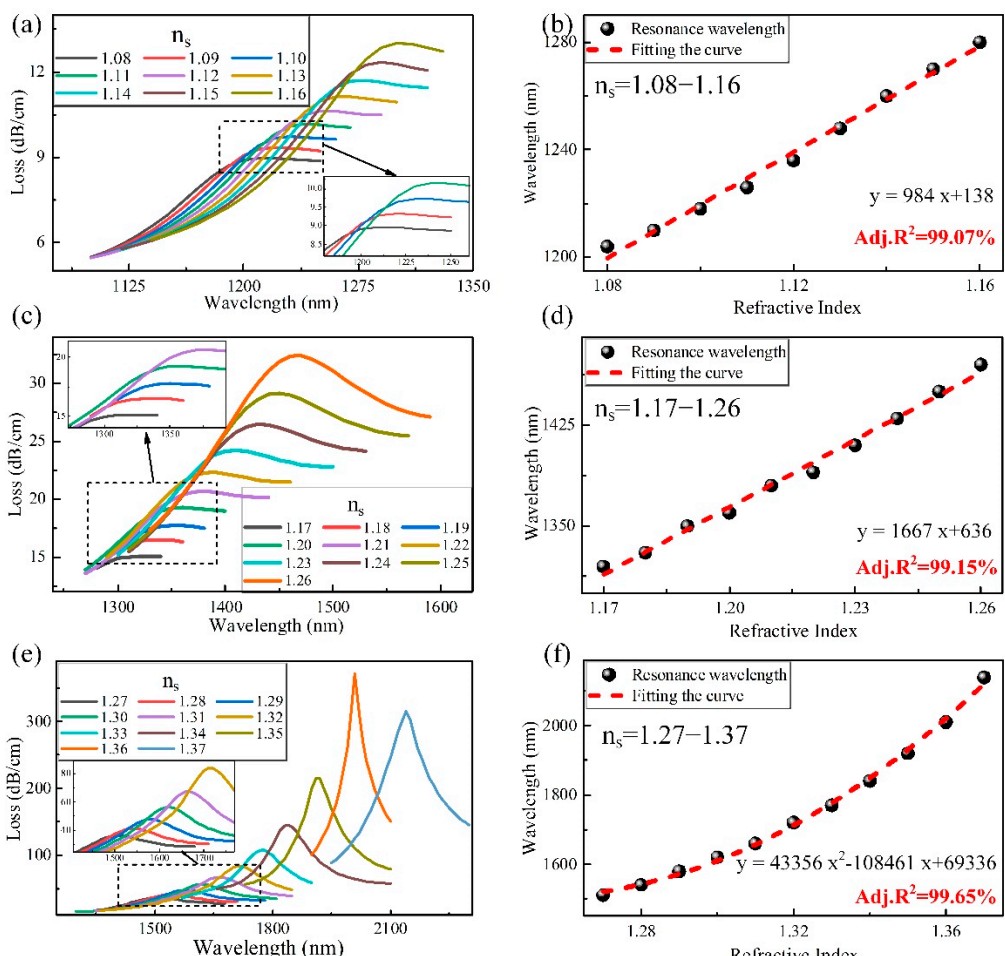

**Figure 7.** (**a**) Loss curves for analytes with refractive indices of 1.08 to 1.16; (**b**) resonance fitting curves for analytes with refractive indices from 1.08 to 1.16; (**c**) loss curves for analytes with refractive indices of 1.17 to 1.26; (**d**) resonance fitting curves for analytes with refractive indices from 1.17 to 1.26; (**e**) loss curves for analytes with refractive indices of 1.27 to 1.37; (**f**) resonance fitting curves for analytes with refractive indices from 1.27 to 1.37.

To illustrate the proposed sensor more strongly, we compare it with the recently proposed sensors by other researchers' recently proposed sensors. The detailed comparison data are shown in Table 2.

**Table 2.** Performance analysis of the same type of sensors.

| Design | $n_s$ Range | Wavelength Range (nm) | Max WS (nm/RIU) | Ave WS (nm/RIU) |
|---|---|---|---|---|
| Au nanowires [20] | 1.33–1.36 | 599–768 | 9000 | 5500 |
| Au nanowires, Low RI [38] | 1.27–1.36 | 640–960 | 9100 | 2350 |
| Ag nanowire, Broad Range [39] | 1.33–1.42 | 530–920 | 10,300 | 4300 |
| Ag nanowire [25] | 1.33–1.38 | 1097–1570 | 9314 | 4730 |
| Low RI, Broad Range, PSPR, LSPR | 1.08–1.16 | 1210–1300 | 2000 | 1125 |
| | 1.17–1.27 | 1320–1510 | 4000 | 1900 |
| | 1.28–1.37 | 1540–2140 | 13,000 | 6667 |

The fabrication of the sensor proposed in this paper is not complicated. Firstly, we selected pure silicon rods of different diameters and made some of them into hollow silica capillaries, then stacked them together in a circle and stretched them at high temperature. Next, a detection plane was polished on one side of the stretched fiber; then a $TiO_2$ layer was coated on that plane; and finally an array of Au nanowires was fabricated by combining

the DC magnetron sputtering, while dip coating and self-assembly evaporation techniques were transferred to the sensing plane [40,41]. Our proposed Au nanowire group array structure is well applied in the field of bioinformatics detection. Besides, the effect of the structure on SPR can also be applied to solar cells [42], photoelectric conversion [43], photoelectric switches [44], and other fields, and it has good application value.

The analysis of manufacturing tolerances allows a good analysis of the errors caused by manufacturing process differences on the structure, which can reflect the stability and feasibility of the device. General manufacturing tolerances are considered in the range of $\pm2\%$ to $\pm10\%$ [26,28]. As shown in Figure 8, we analyze the $\pm5\%$ manufacturing tolerance of the proposed sensor. Figure 8a shows the analysis of the Au nanowire diameter; when d varies around 70 nm, the resonance wavelength changes by 6 nm, and the resonance loss changes by 1 dB/cm. Figure 8b shows the analysis of the Au nanowire group spacing; when $\Lambda_1$ varies around 70 nm, the resonance wavelength changes by 9 nm, and the resonance loss changes by 0.6 dB/cm. Compared with itself, these changes are basically negligible. In conclusion, our designed sensor can effectively reduce the appropriate interference owing to the manufacturing process.

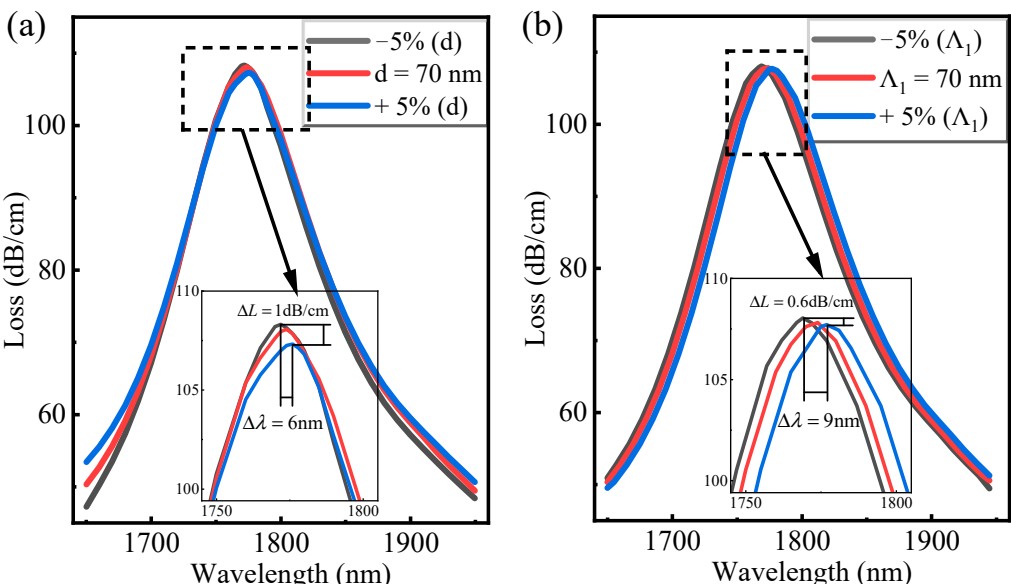

**Figure 8.** Manufacturing tolerance analysis. Analysis of (**a**) Au nanowires diameter and (**b**) straight spacing of Au nanowire groups.

## 4. Conclusions

A PCF sensor based on an array of Au nanowire groups for low RI detection is described and its sensing characteristics are systematically investigated. The Au nanowires are arranged into groups, and then the Au nanowire groups are arranged into an array and placed in the sensing plane of the D-type sensor. The sensing characteristics of the Au nanowire group array with PSPR and LSPR are systematically explained in terms of mechanism, simulation, and numerical analysis, and the detection characteristics of the proposed sensor are more detailed and accurate. The maximum sensitivity of the sensor is 13,000 nm/RIU in the RI range of 1.08 to 1.37, achieving high sensitivity detection in a wide range of the low RI.

**Author Contributions:** G.X. and H.Y. conceived the design; J.S. performed the simulation; J.S. analyzed the simulation data; H.Y. wrote the paper; Z.O., H.L., X.L., Z.C., Y.L. and J.L. made revisions and finalized the paper. All authors have read and agreed to the published version of the manuscript.

**Funding:** This work was supported in part by the National Natural Science Foundation of China (62165004, 61765004); the Innovation Project of GUET Graduate Education (2021YCXS131, 2022YCXS047, 2021YCXS040); and the Open Fund of Foshan University, Research Fund of Guangdong Hong Kong-Macao Joint Laboratory for Intelligent Micro-Nano Optoelectronic Technology (2020B1212030010).

**Institutional Review Board Statement:** Not applicable.

**Informed Consent Statement:** Not applicable.

**Data Availability Statement:** Data sharing not applicable.

**Conflicts of Interest:** The authors declare no conflict of interest.

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
