# Peer review of "Fiber Optic Sensor with a Gold Nanowire Group Array for Broad Range and Low Refractive Index Detection"

_photonics, doi:10.3390/photonics9090661_

Round 1

Reviewer 1 Report

In this manuscript, an ultra-wide range high sensitivity plasmonic fiber optic sensor based on a D-shaped photonic crystal fiber with a nanowire group array was proposed. Benefit from the simultaneous generation of PSPR and LSPR, the sensitivity and the measurement range were significantly improved compared to sensors of the same type. The manuscript has been well written and has a high degree of innovation, and I am happy to recommend it for publication. There are some points should be addressed prior to publication, as detailed below.

1.        The authors should explain why the RW in Table 1 are all multiples of 10, which can lead to huge errors in the calculated WS.

2.        Why does the Loss decrease rapidly when the wavelength is less than the resonance wavelength, and when the wavelength exceeds the resonance wavelength, the Loss remains almost unchanged, and even continues to rise (n=1.17, 1.18)?

3.        Moreover, the data in Figures 7 do not correspond to Table 1 at all. For example, when n=1.17, the RW given in Table 1 is 1320 nm, but the calculation for 1320 nm is not performed in Figure 7c, and the corresponding point in Figure 7d is less than 1300 nm. The Loss in Table 1 is 13.75 dB/cm, however in Figure 7c it clearly exceeds 15 dB/cm.

4.        In addition, this article has many formatting issues: for example, the PML in Figure 1 and the SPP on line 150 are not given full writing; "detectors" in line 155 should be changed to "analytes"; 104 should not be multiplied in Equation 2. The authors should seriously modify the whole manuscript.

Author Response

Response to Reviewer 1 Comments

Manuscript ID: 1900449

Title: Fiber optic sensor with gold nanowire groups array for broad range and low refractive index detection

Dear Editors and Reviewers:

Thank you for your letter and for the reviewers’ comments concerning our manuscript entitled “Fiber optic sensor with gold nanowire groups array for broad range and low refractive index detection” (ID: 1900449).

Those comments are all valuable and very helpful for revising and improving our paper, as well as the important guiding significance to our researches. We have studied comments carefully and have made correction which we hope meet with approval. Revised portion are marked in red in the paper. The main corrections in the paper and the responds to the reviewer’s comments are as flowing. We appreciate for Editors/Reviewers’ warm work earnestly, and hope that the correction will meet with approval.

Once again, thank you very much for your comments and suggestions.

Response to Reviewer 1 Comments

Comments: In this manuscript, an ultra-wide range high sensitivity plasmonic fiber optic sensor based on a D-shaped photonic crystal fiber with a nanowire group array was proposed. Benefit from the simultaneous generation of PSPR and LSPR, the sensitivity and the measurement range were significantly improved compared to sensors of the same type. The manuscript has been well written and has a high degree of innovation, and I am happy to recommend it for publication. There are some points should be addressed prior to publication, as detailed below.

Point 1: The authors should explain why the RW in Table 1 are all multiples of 10, which can lead to huge errors in the calculated WS.

Response 1: We thank the reviewers for their comments. In Table 1, the minimum accuracy of resonance wavelength is 10 nm, which is governed by both the accuracy of sensitivity and the accuracy of simulation. If the minimum accuracy of the resonance wavelength is too large, the accuracy of the sensitivity will be reduced; if the minimum accuracy of the resonance wavelength is too small, the simulation difficulty increases. In order to ensure both, the minimum accuracy of resonance wavelength is therefore set at 10 nm, which is also recognized in the same type of articles [1,2]. However, it is undeniable that the minimum accuracy of 10 nm may indeed produce errors in the calculation of wavelength sensitivity at low refractive indices, the authors will optimize the accuracy reasonably in subsequent studies to reduce such errors.

  1. Asaduzzaman, S.; Ahmed, K. Investigation of ultra-low loss surface plasmon resonance-based PCF for biosensing application. Results in Physics 2018, 11, 358-361, doi:10.1016/j.rinp.2018.09.026.
  2. Al Noman, A.; Haque, E.; Hossain, M.A.; Nguyen Hoang, H.; Namihira, Y.; Ahmed, F. Sensitivity Enhancement of Modified D-Shaped Microchannel PCF-Based Surface Plasmon Resonance Sensor. Sensors 2020, 20, doi:10.3390/s20216049.

Point 2: Why does the Loss decrease rapidly when the wavelength is less than the resonance wavelength, and when the wavelength exceeds the resonance wavelength, the Loss remains almost unchanged, and even continues to rise (n=1.17, 1.18)?

Response 2: We thank the reviewers for their comments. The loss is generated by the strong coupling formed after the phase matching of core mode and SPP mode. At this time the energy in the fiber core flows to the transmission plane and thus the loss becomes large. Under normal circumstances, the loss not near the resonant wavelength is relatively small. Phase matching point, however, is not only one in the analysis of the refractive index is constant, the loss may have more than one peak. It formed the crest on the left side of the change is greater than the right side of the peak, when there is multiple loss peak, on the right side of the loss of the first peak loss will slowly drop, reached a nadir slowly rising, when close to the loss of the second peak resonance wavelength, the loss curve goes up sharply again. The authors reviewed the relevant literature and the phenomenon was confirmed [1,2].

  1. Wang, F.; Liu, C.; Sun, Z.; Sun, T.; Liu, B.; Chu, P.K. A Highly Sensitive SPR Sensors Based on Two Parallel PCFs for Low Refractive Index Detection. Ieee Photonics Journal 2018, 10, doi:10.1109/jphot.2018.2856273.
  2. Liu, C.; Su, W.; Wang, F.; Li, X.; Liu, Q.; Mu, H.; Sun, T.; Chu, P.K.; Liu, B. Birefringent PCF-Based SPR Sensor for a Broad Range of Low Refractive Index Detection. Ieee Photonics Technology Letters 2018, 30, 1471-1474, doi:10.1109/lpt.2018.2856859.

Point 3: Moreover, the data in Figures 7 do not correspond to Table 1 at all. For example, when n=1.17, the RW given in Table 1 is 1320 nm, but the calculation for 1320 nm is not performed in Figure 7c, and the corresponding point in Figure 7d is less than 1300 nm. The Loss in Table 1 is 13.75 dB/cm, however in Figure 7c it clearly exceeds 15 dB/cm.

Response 3: Thank you for your valuable comments. The reviewers have analyzed the data in Figure 7 and Table 1 and found out the unreasonable data. In this regard, the authors are very grateful to the reviewer for bringing up the errors in the article and ensuring the rigor of the article. The authors have re-verified the simulation and data and found that some of the data were incorrectly plotted. The authors have checked all data in this section and corrected the erroneous data. Figure 7 now matches the data in Table 1.

Figure 7. (a) Loss curves for analytes with refractive indices of 1.08 to 1.16; (b) Resonance fitting curves for analytes with refractive indices from 1.08 to 1.16; (c) Loss curves for analytes with refractive indices of 1.17 to 1.26; (d) Resonance fitting curves for analytes with refractive indices from 1.17 to 1.26; (e) Loss curves for analytes with refractive indices of 1.27 to 1.37; (f) Resonance fitting curves for analytes with refractive indices from 1.27 to 1.37.

Table 1. Performance analysis of sensor.

ns

RW

(nm)

Loss

(dB/cm)

WS

(nm/RIU)

ns

RW

(nm)

Loss

(dB/cm)

WS

(nm/RIU)

1.08

1210

8.97

1000

1.23

1410

24.22

2000

1.09

1220

9.33

1000

1.24

1430

26.46

2000

1.10

1230

9.73

1000

1.25

1450

29.04

2000

1.11

1240

10.16

1000

1.26

1470

32.37

4000

1.12

1250

10.62

2000

1.27

1510

36.35

3000

1.13

1270

11.34

1000

1.28

1540

41.33

4000

1.14

1280

11.71

1000

1.29

1580

47.72

4000

1.15

1290

12.33

1000

1.30

1620

56.17

4000

1.16

1300

13.34

2000

1.31

1660

67.51

6000

1.17

1320

15.01

1000

1.32

1720

83.63

5000

1.18

1330

16.48

2000

1.33

1770

107.62

7000

1.19

1350

17.71

1000

1.34

1840

145.64

8000

1.20

1360

19.06

2000

1.35

1920

215.12

9000

1.21

1380

20.60

1000

1.36

2010

371.76

13000

1.22

1390

22.30

2000

1.37

2140

315.63

NA

In lines 267 to 271, “When the refractive index range is 1.17 to 1.26, the maximum WS is 4000 nm/RIU, the average WS is 2000 nm/RIU, and the fitted curve of resonance wavelength is , and the adjusted R2 is 99.79%.” modified to “When the refractive index range is 1.17 to 1.26, the maximum WS is 4000 nm/RIU, the average WS is 2000 nm/RIU, and the fitting curve of resonance wavelength is , and the adjusted R2 is 99.15%.”

Point 4: In addition, this article has many formatting issues: for example, the PML in Figure 1 and the SPP on line 150 are not given full writing; "detectors" in line 155 should be changed to "analytes"; 104 should not be multiplied in Equation 2. The authors should seriously modify the whole manuscript.

Response 4: We thank the reviewers for their valuable suggestions. The authors have re-reviewed the article and corrected errors in it. In response to the questions raised by the reviewers in this comment, the authors will answer them below.

In lines 106-107, additional content “The outermost layer of the fiber is an ideal cladding, which we can consider as a perfect match layer (PML).”

In lines 112-113, “the dielectric constants of Au and Ag can be expressed in the Drude-Lorenz model,” modified to “the dielectric constants of Au can be expressed in the Drude-Lorenz model,”

In line 174, “this mode is called SPP mode,” modified to “this mode is called surface plasmon polaritons (SPP) mode,”

In lines 182-183, “Different detectors have different refractive indices,” modified to “Different analytes have different refractive indices,”.

In line 119, “ ” modified to “ ”.

Reviewer 2 Report

The article reported by Xiao et al. shows a numerical investigation over plasmonic refractive index sensors utilising multiple Au nanowires. The article is purely numerical analysis but has some potential, however, there are some open points that need to be addressed before any consideration. Comments are given below:

1. State of the art needs to be improved.  There are several articles published in recent years based on similar technology utilising nanowires such as 10.3390/s19173730, 10.1109/LPT.2022.3182783, 10.1364/OME.8.003927, 10.1109/JPHOT.2021.3069396, 10.1109/LPT.2020.2980470, hence I recommend authors improve the state of the art around nanowires and their progress instead of thin or nanofilms (e.g., Ref. 13, 14, 15) and then discuss the advantage of the reported sensor and how it is taking the filed forward. Since it's pure theoretical work therefore they can't consider the performance with some experimental work.

2. Similarly Table 2 must be changed by comparing the data and sensing performance with similar technology using nanowires instead of thin films.

3. Fig. 3 Authors are advised to mention the core mode, SPP mode, and phase-matched mode separately.

4. Why there is a small kink in core mode and spp mode in the dispersion curve? SPP modes are highly lossy and decrease sharply but it looks like it shows a small kink before 1800, similarly, the core more (correct typo in the graph) shows a kink after 1800 which means something happens during the simulation. Authors are advised to discuss the reason for this behavior.

5. At the phase matching point the energy is completely transferred from core mode to SPP mode and gives the maximum modal loss while as I can see from Fig. 3, the maximum loss is obtained far away from the phase matching point, why so? reason must be discussed.

6. Fabrication tolerance must be included in the result and discussion for each parameter.

7. Dispersed typos and grammatical errors.

Author Response

Response to Reviewer 2 Comments

Manuscript ID: 1900449

Title: Fiber optic sensor with gold nanowire groups array for broad range and low refractive index detection

Dear Editors and Reviewers:

Thank you for your letter and for the reviewers’ comments concerning our manuscript entitled “Fiber optic sensor with gold nanowire groups array for broad range and low refractive index detection” (ID: 1900449).

Those comments are all valuable and very helpful for revising and improving our paper, as well as the important guiding significance to our researches. We have studied comments carefully and have made correction which we hope meet with approval. Revised portion are marked in red in the paper. The main corrections in the paper and the responds to the reviewer’s comments are as flowing. We appreciate for Editors/Reviewers’ warm work earnestly, and hope that the correction will meet with approval.

Once again, thank you very much for your comments and suggestions.

Response to Reviewer 2 Comments

Comments: The article reported by Xiao et al. shows a numerical investigation over plasmonic refractive index sensors utilising multiple Au nanowires. The article is purely numerical analysis but has some potential, however, there are some open points that need to be addressed before any consideration. Comments are given below:

Point 1: State of the art needs to be improved. There are several articles published in recent years based on similar technology utilising nanowires such as 10.3390/s19173730, 10.1109/LPT.2022.3182783, 10.1364/OME.8.003927, 10.1109/JPHOT.2021.3069396, 10.1109/LPT.2020.2980470, hence I recommend authors improve the state of the art around nanowires and their progress instead of thin or nanofilms (e.g., Ref. 13, 14, 15) and then discuss the advantage of the reported sensor and how it is taking the filed forward. Since it's pure theoretical work therefore they can't consider the performance with some experimental work.

Response 1: We thank the reviewers for their valuable suggestions and relevant literature. The main idea of the article is to demonstrate the superiority of the proposed gold nanowire groups array fiber optic sensor by performing a comparative analysis with the conventional PSPR and LSPR sensors. In the state-of-the-art section, the authors describe the PSPR sensor and the LSPR sensor, respectively. After receiving this reviewer's comments, the authors did find the description of the type of gold nanowire sensors to be inadequate. However, the authors believe that the description of thin-film sensors, although not the focus, is still necessary to exist. Therefore, in the state-of-the-art section, the authors reduced the description about thin film sensors and added the study about metal nanowire sensors. By analyzing their advantages and disadvantages to better discuss the advantages of the designed sensors and how to move forward.

In lines 70-87, additional content “In recent years, with the improvement of manufacturing process, more and more complex structures have been applied in the field of sensing and detection, among which the proposed metal nanowire arrays provide new research ideas for optical fiber sensors [1,2]. The metal nanowire array actually belongs to LSPR, but it is different from it in that the metal nanowire array is composed of multiple metal wires, which may have new detection properties along with the traditional LSPR properties. Pathak, A.K. et al. proposed a highly sensitive bio-detection by embedding metal nanowires into a dual-hole microchannel. The design utilizes two Au nanowires to excite two modes of coupling to achieve high performance detection characteristics with a sensitivity of 90500 nm/RIU at a refractive index of 1.40. Meshginqalam, B. and Barvestani, J. proposed a highly sensitive fiber optic sensor for cancer cell detection using two sets of bimetallic wires. The design placed two sets of bimetallic wires at the upper and lower ends of the optical fiber and obtained good detection performance after rational optimization of the structural parameters, and this sensor can be used to detect six different cancer cells. Zhan, Y. et al. designed a fiber optic sensor based on metal nanowire surrounds, which replaced the traditional metal thin layer with a ring-shaped metal nanowire, which improved the detection performance of the sensor. Sensor has potential applications in biochemical detection. In conclusion, metal nanowire arrays have good research value in the field of detection.”

  1. Pathak, A.K.; Singh, V.K. SPR Based Optical Fiber Refractive Index Sensor Using Silver Nanowire Assisted CSMFC. Ieee Photonics Technology Letters 2020, 32, 465-468, doi:10.1109/lpt.2020.2980470.
  2. Zhao, L.; Han, H.; Luan, N.; Liu, J.; Song, L.; Hu, Y. A Temperature Plasmonic Sensor Based on a Side Opening Hollow Fiber Filled with High Refractive Index Sensing Medium. Sensors 2019, 19, doi:10.3390/s19173730.
  3. Pathak, A.K.; Rahman, B.M.A.; Viphavakit, C. Nanowire Embedded Micro-Drilled Dual-Channel Approach to Develop Highly Sensitive Biosensor. Ieee Photonics Technology Letters 2022, 34, 707-710, doi:10.1109/lpt.2022.3182783.
  4. Meshginqalam, B.; Barvestani, J. High performance surface plasmon resonance-based photonic crystal fiber biosensor for cancer cells detection. European Physical Journal Plus 2022, 137, doi:10.1140/epjp/s13360-022-02618-6.
  5. Zhan, Y.; Li, Y.; Wu, Z.; Hu, S.; Li, Z.; Liu, X.; Yu, J.; Huang, Y.; Jing, G.; Lu, H.; et al. Surface plasmon resonance-based microfiber sensor with enhanced sensitivity by Au nanowires. Optical Materials Express 2018, 8, 3927-3940, doi:10.1364/ome.8.003927.

Point 2: Similarly, Table 2 must be changed by comparing the data and sensing performance with similar technology using nanowires instead of thin films.

Response 2: We thank the reviewers for their suggestions. In Table2, the comparative analysis of the data for the metal nanowire sensors is indeed missing. The authors have reviewed the relevant literature and revised and added to Table 2.

Table 2. Performance analysis of the same type of sensors.

Design

ns range

Wavelength range (nm)

Max WS

(nm/RIU)

Ave.WS

(nm/RIU)

Low RI, Broad Range

1.20-1.40

1260-1570

3752

1550

Low RI, Broad Rang

1.25-1.43

710-1150

6800

2444

Low RI, Broad Range

1.00-1.60

790-1410

2275

1033

Low RI

1.29-1.36

1220-1662

9245

6457

Au nanowires [1]

1.33-1.36

599-768

9000

5500

Au nanowires Low RI [2]

1.27-1.36

640-960

9100

2350

Low RI, Broad Range,

PSPR, LSPR

1.08-1.16

1210-1300

2000

1125

1.17-1.27

1320-1510

4000

1900

1.28-1.37

1540-2140

13000

6667

  1. Jiao, S.; Gu, S.; Yang, H.; Fang, H. Research on dual-core photonic crystal fiber based on local surface plasmon resonance sensor with silver nanowires. Journal of Nanophotonics 2018, 12, doi:10.1117/1.Jnp.12.046015.
  2. Liu, C.; Yang, L.; Liu, Q.; Wang, F.; Sun, Z.; Sun, T.; Mu, H.; Chu, P.K. Analysis of a Surface Plasmon Resonance Probe Based on Photonic Crystal Fibers for Low Refractive Index Detection. Plasmonics 2018, 13, 779-784, doi:10.1007/s11468-017-0572-7.

Point 3: Fig. 3 Authors are advised to mention the core mode, SPP mode, and phase-matched mode separately.

Response 3: We thank the reviewers for their suggestions. The authors' check revealed a real problem with the presentation of the model in Figure 3. The authors have made the appropriate changes as suggested.

In Fig. 3, ‘Coro mode, SPP mode, Resonance mode’ modified to ‘Core mode, SPP mode, Phase-matched mode’.

Point 4: Why there is a small kink in core mode and spp mode in the dispersion curve? SPP modes are highly lossy and decrease sharply but it looks like it shows a small kink before 1800, similarly, the core more (correct typo in the graph) shows a kink after 1800 which means something happens during the simulation. Authors are advised to discuss the reason for this behavior.

Response 4: We thank the reviewers for their comments. The reviewer's query about the small kink in Figure 3 is answered by the author after consulting relevant information and taking into account the actual situation of his own design [1,2]. The two blue curves in Figure 3 are the real parts of the effective refractive indices of the core mode and the SPP mode. The SPP mode is a sharp drop and the core mode is a slow drop. When the incident light is a certain wavelength, these two modes will be strongly coupled and most of the energy in the core will flow to the detection plane, which is the resonance wavelength. Near the resonance wavelength, the real part of the effective refractive index of the two modes will change because they are coupled, resulting in a small kink. In Figure 3, the resonance wavelength is about 1800 nm, and after analyzing the simulation and data, it is found that the different kink positions of the two modes are caused by the low accuracy of the simulation and the accuracy of the plot. The authors re-performed the simulation and plotting of Figure 3.

  1. Wang, G.; Li, S.; An, G.; Wang, X.; Zhao, Y.; Zhang, W.; Chen, H. Highly sensitive D-shaped photonic crystal fiber biological sensors based on surface plasmon resonance. Optical and Quantum Electronics 2016, 48, doi:10.1007/s11082-015-0346-4.
  2. Wu, T.; Shao, Y.; Wang, Y.; Cao, S.; Cao, W.; Zhang, F.; Liao, C.; He, J.; Huang, Y.; Hou, M.; et al. Surface plasmon resonance biosensor based on gold-coated side-polished hexagonal structure photonic crystal fiber. Optics Express 2017, 25, 20313-20322, doi:10.1364/oe.25.020313.

In line 176 to 180, “When the real parts of these two modes are equal, a resonant mode occurs, which is represented by a loss curve.” modified to “When the incident light is a certain value, the two modes will be phase matched to produce a strong coupling, and the real part of the two modes, which was originally monotonically decreasing, produces a small kink near this wavelength. At the same time, a large amount of energy in the core flows to the detection plane. The wavelength that can produce these phenomena is called the resonance wavelength.”

Point 5: At the phase matching point the energy is completely transferred from core mode to SPP mode and gives the maximum modal loss while as I can see from Fig. 3, the maximum loss is obtained far away from the phase matching point, why so? reason must be discussed.

Response 5: Thank you for your comments. The authors believe that there are two reasons why the wavelength at the peak of the loss curve in Figure 3 is different from the wavelength at the intersection of the core mode and the SPP mode. The first point is the problem of simulation accuracy and plotting accuracy. The authors used a higher accuracy for the simulation and plotting, and the new figure can be seen in the answer to question 4, where the wavelengths are quite close to each other. The second point is that the loss curve refers to all the loss in the fiber, the resonance phenomenon in the detection plane occupies most of the loss, but there is still a very small part of other loss, which causes a very small deviation in the two wavelengths, because this deviation is very small, so we can consider that the two wavelengths are equal.

Point 6: Fabrication tolerance must be included in the result and discussion for each parameter.

Response 6: Thank you for your valuable comments. It is true that the authors did not consider the manufacturing tolerance. In this design, the diameter of the gold nanowires and the spacing between the gold nanowire groups have a large impact on the detection performance, and after reviewing the relevant literature, a manufacturing tolerance analysis of ±5% for these two parameters was added.

Figure 8. Manufacturing tolerance analysis. Analysis of (a) Au nanowires diameter and (b) straight spacing of Au nanowire groups.

In lines 296-306, additional content“The analysis of manufacturing tolerances allows a good analysis of the errors caused by manufacturing process differences on the structure, which can reflect the stability and feasibility of the device. General manufacturing tolerances are considered in the range of 2% to 10% [1,2]. As shown in Figure 8, we analyze the ±5% manufacturing tolerance of the proposed sensor. Figure 8(a) shows the analysis of the Au nanowire diameter, when D varies around 70 nm, the resonance wavelength changes by 6 nm, and the resonance loss changes by 1 dB/cm; Figure 8(b) shows the analysis of the Au nanowire group spacing, when É…1 varies around 70 nm, the resonance wavelength changes by 9 nm, and the resonance loss changes by 0.6 dB/cm. Compared with itself, these changes are basically negligible. In conclusion, our designed sensor can effectively reduce the appropriate interference due to the manufacturing process. ”

  1. Pathak, A.K.; Rahman, B.M.A.; Viphavakit, C. Nanowire Embedded Micro-Drilled Dual-Channel Approach to Develop Highly Sensitive Biosensor. Ieee Photonics Technology Letters 2022, 34, 707-710, doi:10.1109/lpt.2022.3182783.
  2. Zhao, L.; Han, H.; Luan, N.; Liu, J.; Song, L.; Hu, Y. A Temperature Plasmonic Sensor Based on a Side Opening Hollow Fiber Filled with High Refractive Index Sensing Medium. Sensors 2019, 19, doi:10.3390/s19173730.

Point 7: Dispersed typos and grammatical errors.

Response 7: We thank the reviewers for their valuable suggestions. The authors have re-reviewed the article and corrected the mistakes therein. Changes made are placed below.

In lines 106-107, additional content “The outermost layer of the fiber is an ideal cladding, which we can consider as a perfect match layer (PML).”

In lines 112-113, “the dielectric constants of Au and Ag can be expressed in the Drude-Lorenz model,” modified to “the dielectric constants of Au can be expressed in the Drude-Lorenz model,”

In line 174, “this mode is called SPP mode,” modified to “this mode is called surface plasmon polaritons (SPP) mode,”

In lines 182-183, “Different detectors have different refractive indices,” modified to “Different analytes have different refractive indices,”.

In line 119, “ ” modified to “ ”.

Reviewer 3 Report

In this manuscript, the authors designed an optical fiber sensor based on LSPR and PSPR, which utilizes the special detection properties of gold nanowire groups array to achieve a wide range of detection at low refractive indices. The authors have well verified the detection performance of the sensor by mechanism analysis, simulation analysis and numerical analysis. Overall, this work is interesting and supported by adequate data and novelty. I suggest that this manuscript could be accepted with minor revision.

(1)  Did the author consider the quantum tunneling effect between metal wires when designing the metal nanowire group? If this effect exists, please do the analysis.

(2)  In this paper, the authors present a theoretical and simulation analysis of the proposed sensor, but do not analyze its feasibility. Please add the feasibility analysis of this sensor.

(3)  The English of the paper needs improvement. There are some linguistic errors in the text.

(4)  Please explain how the sensing characteristics of LSPR and PSPR proposed by the author can be used in other fields.

(5)  Please add some relevant literature to increase the credibility and accuracy of the article.

Author Response

Response to Reviewer 3 Comments

Manuscript ID: 1900449

Title: Fiber optic sensor with gold nanowire groups array for broad range and low refractive index detection

Dear Editors and Reviewers:

Thank you for your letter and for the reviewers’ comments concerning our manuscript entitled “Fiber optic sensor with gold nanowire groups array for broad range and low refractive index detection” (ID: 1900449).

Those comments are all valuable and very helpful for revising and improving our paper, as well as the important guiding significance to our researches. We have studied comments carefully and have made correction which we hope meet with approval. Revised portion are marked in red in the paper. The main corrections in the paper and the responds to the reviewer’s comments are as flowing. We appreciate for Editors/Reviewers’ warm work earnestly, and hope that the correction will meet with approval.

Once again, thank you very much for your comments and suggestions.

Response to Reviewer 3 Comments

Comments: In this manuscript, the authors designed an optical fiber sensor based on LSPR and PSPR, which utilizes the special detection properties of gold nanowire groups array to achieve a wide range of detection at low refractive indices. The authors have well verified the detection performance of the sensor by mechanism analysis, simulation analysis and numerical analysis. Overall, this work is interesting and supported by adequate data and novelty. I suggest that this manuscript could be accepted with minor revision.

Point 1: Did the author consider the quantum tunneling effect between metal wires when designing the metal nanowire group? If this effect exists, please do the analysis.

Response 1: We thank the reviewers for their valuable comments. In the gold nanowire group arrays designed by the authors, the gold nanowires in each group are in contact with each other. The contact area between adjacent gold nanowires is small from the geometrical analysis, but the electric field strength at the connection and the displacement current density in the x-direction are large as shown in Figure 4. After receiving the comments from the reviewers, the authors reviewed the relevant information and reanalyzed the simulation model, and found that there is indeed a quantum tunneling effect between the adjacent gold nanowires. Near the contact point, one is due to the very close distance of the two metal nanowires, and the other is due to the large field strength here, the electrons in the adjacent gold nanowires achieve a small mutual movement without contact, which is consistent with the quantum tunneling effect. According to this effect, the phenomenon near the connection in Figure 4 can be well explained. The authors have added relevant contents to the article.

In lines 205-210, additional content“Observing Figure 4 (b) and (c) we can find that there is a large electric field and displacement current density at the contact points of the adjacent Au nanowires. This is because not only there is EW controlling the charge movement at the contact point, but also near the contact point, due to the excessive high energy and very small distance, there may be quantum tunneling effect between the two Au nanowires, which in turn increases the charge flow between the Au nanowires [1,2].”

  1. Hooshmand, N.; Mousavi, H.S.; Panikkanvalappil, S.R.; Adibi, A.; El-Sayed, M.A. High-sensitivity molecular sensing using plasmonic nanocube chains in classical and quantum coupling regimes. Nano Today 2017, 17, 14-22, doi:10.1016/j.nantod.2017.10.009.
  2. Liu, D.; Wu, T.; Zhang, Q.; Wang, X.; Guo, X.; Su, Y.; Zhu, Y.; Shao, M.; Chen, H.; Luo, Y.; et al. Probing the in-Plane Near-Field Enhancement Limit in a Plasmonic Particle-on-Film Nanocavity with Surface-Enhanced Raman Spectroscopy of Graphene. Acs Nano 2019, 13, 7644-7654, doi:10.1021/acsnano.9b00776.

Point 2: In this paper, the authors present a theoretical and simulation analysis of the proposed sensor, but do not analyze its feasibility. Please add the feasibility analysis of this sensor.

Response 2: We thank the reviewers for their comments. The authors analyzed the proposed sensor model from theoretical, simulation and numerical perspectives, and the analysis in terms of manufacturing feasibility is really missing. The authors reviewed the literature and combined their own sensor characteristics to analyze the manufacturing feasibility.

In lines 286-292, additional content“The fabrication of the sensor proposed in this paper is not complicated. Firstly, we selected pure silicon rods of different diameters and made some of them into hollow silica capillaries, then stacked them together in a circle and stretched them at high temperature. Next, a detection plane was polished on one side of the stretched fiber, then a TiO2 layer was coated on that plane, and finally, an array of Au nanowires array fabricated by combining the DC magnetron sputtering, dip coating and self-assembly evaporation techniques were transferred to the sensing plane [1,2]. ”

  1. Wieduwilt, T.; Kirsch, K.; Dellith, J.; Willsch, R.; Bartelt, H. Optical Fiber Micro-Taper with Circular Symmetric Gold Coating for Sensor Applications Based on Surface Plasmon Resonance. Plasmonics 2013, 8, 545-554, doi:10.1007/s11468-012-9432-7.
  2. Jing, G.; Bodiguel, H.; Doumenc, F.; Sultan, E.; Guerrier, B. Drying of Colloidal Suspensions and Polymer Solutions near the Contact Line: Deposit Thickness at Low Capillary Number. Langmuir 2010, 26, 2288-2293, doi:10.1021/la9027223.

Point 3: The English of the paper needs improvement. There are some linguistic errors in the text.

Response 3: We thank the reviewers for their valuable suggestions. The authors have re-reviewed the article and corrected the mistakes therein. Changes made are placed below.

In lines 106-107, additional content “The outermost layer of the fiber is an ideal cladding, which we can consider as a perfect match layer (PML).”

In lines 112-113, “the dielectric constants of Au and Ag can be expressed in the Drude-Lorenz model,” modified to “the dielectric constants of Au can be expressed in the Drude-Lorenz model,”

In line 174, “this mode is called SPP mode,” modified to “this mode is called surface plasmon polaritons (SPP) mode,”

In lines 182-183, “Different detectors have different refractive indices,” modified to “Different analytes have different refractive indices,”.

In line 119, “ ” modified to “ ”.

Point 4: Please explain how the sensing characteristics of LSPR and PSPR proposed by the author can be used in other fields.

Response 4: We thank the reviewers for their valuable suggestions. We designed the gold nanowire group array with both PSPR and LSPR properties. SPR can be used not only as sensing, but also in other fields, such as solar cells [1], photoelectric conversion [2], photoelectric switches [3], so as long as the gold nanowire group array is reasonably combined with related devices, it will certainly have good applications in other fields.

In lines 292-295, additional content“Our proposed Au nanowire group array structure is well applied in the field of bioinformatics detection. Besides, the effect of the structure on SPR can also be applied to solar cells [1], photoelectric conversion [2], photoelectric switches [3] and other fields, and its has good application value. ”

  1. Zhang, J.-C.; Mao, Y.; Wang, W.; Guan, Y.-R.; Bao, Y.; Niu, L. In Situ Investigation on Electrochemical Polymerization and Properties of Polyaniline Thin Films by Electrochemical Surface Plasmon Resonance. Chinese Journal of Analytical Chemistry 2015, 43, 350-355.
  2. Tan, D.; Wang, Y.; Gan, Y. Facile Visible-Light-Assisted Synthesis, Optical, and Electrochemical Properties of Pd Nanoparticles with Single crystalline and Multiple-twinned Structures. Rare Metal Materials and Engineering 2017, 46, 2065-2069.
  3. Baba, A.; Tada, K.; Janmanee, R.; Sriwichai, S.; Shinbo, K.; Kato, K.; Kaneko, F.; Phanichphant, S. Controlling Surface Plasmon Optical Transmission with an Electrochemical Switch Using Conducting Polymer Thin Films. Advanced Functional Materials 2012, 22, 4383-4388, doi:10.1002/adfm.201200373.

Point 5: Please add some relevant literature to increase the credibility and accuracy of the article.

Response 5: We thank the reviewers for their valuable suggestions. The author does lack literature support when describing relevant content. The authors have consulted the relevant literature and supplemented to the content.

In lines 70-87, additional content “In recent years, with the improvement of manufacturing process, more and more complex structures are being applied in the field of sensing and detection, among which the proposed metal nanowire arrays provide new research ideas for optical fiber sensors [1,2]. The metal nanowire array actually belongs to LSPR, but it is different from it in that the metal nanowire array is composed of multiple metal wires, which may have new detection properties along with the traditional LSPR properties. Pathak, A.K. et al. proposed a highly sensitive bio-detection by embedding metal nanowires into a dual-hole microchannel. The design utilizes two Au nanowires to excite two modes of coupling to achieve high performance detection characteristics with a sensitivity of 90500 nm/RIU at a refractive index of 1.40. Meshginqalam, B. and Barvestani, J. proposed a highly sensitive fiber optic sensor for cancer cell detection using two sets of bimetallic wires. The design placed two sets of bimetallic wires at the upper and lower ends of the optical fiber and obtained good detection performance after rational optimization of the structural parameters, and this sensor can be used to detect six different cancer cells. Zhan, Y. et al. designed a fiber optic sensor based on metal nanowire surrounds, which replaced the traditional metal thin layer with a ring-shaped metal nanowire, which improved the detection performance of the sensor. sensor has potential applications in biochemical detection. In conclusion, metal nanowire arrays have good research value in the field of detection.”

  1. Pathak, A.K.; Singh, V.K. SPR Based Optical Fiber Refractive Index Sensor Using Silver Nanowire Assisted CSMFC. Ieee Photonics Technology Letters 2020, 32, 465-468, doi:10.1109/lpt.2020.2980470.
  2. Zhao, L.; Han, H.; Luan, N.; Liu, J.; Song, L.; Hu, Y. A Temperature Plasmonic Sensor Based on a Side Opening Hollow Fiber Filled with High Refractive Index Sensing Medium. Sensors 2019, 19, doi:10.3390/s19173730.
  3. Pathak, A.K.; Rahman, B.M.A.; Viphavakit, C. Nanowire Embedded Micro-Drilled Dual-Channel Approach to Develop Highly Sensitive Biosensor. Ieee Photonics Technology Letters 2022, 34, 707-710, doi:10.1109/lpt.2022.3182783.
  4. Meshginqalam, B.; Barvestani, J. High performance surface plasmon resonance-based photonic crystal fiber biosensor for cancer cells detection. European Physical Journal Plus 2022, 137, doi:10.1140/epjp/s13360-022-02618-6.
  5. Zhan, Y.; Li, Y.; Wu, Z.; Hu, S.; Li, Z.; Liu, X.; Yu, J.; Huang, Y.; Jing, G.; Lu, H.; et al. Surface plasmon resonance-based microfiber sensor with enhanced sensitivity by gold nanowires. Optical Materials Express 2018, 8, 3927-3940, doi:10.1364/ome.8.003927.

Reviewer 4 Report

In this manuscript, the authors studied a gold nanowire-based fiber optic sensor utilizing both PSPR and LSPR. From the simulation results, the proposed fiber optic sensor configuration helps improve the sensitivity of refractive index in the analytes. Overall, this work is thorough and interesting. It could potentially inspire the development of SPR based fiber optic sensors. However, I have the following comments and suggestions that hope authors can help address.

1. In the introduction, it would be helpful if the authors could briefly specify what is the typical range of "low refractive index" and what is the typical definition of "broad range detection".

2. In Line [34], the authors claims "Compared to conventional detection, detection in the low refractive index (RI) range is often neglected by researchers, not only because there are fewer analytes to detect in this range, but also because it is more difficult to detect the low RI range, let alone the broad range detection of low refractive index [1,2]."  This claim is self-contradictory with the content from Line [45] to Line [69] where the authors mentioned reference [11-21] numerous works has been studied in this field in the recent years. Also, if there are "fewer analytes" in this range, the significance of this work will be undermined. I suggest the authors to rephrase the wording here.

3.  In Section 2.2 Principle Analysis, the authors provided an intuitive analysis of why PSPR+LSPR sensor has the potential for better sensing performance. Is it possible to include theoretical study to be more specific and help cross-validate with the simulation study in the following section?

4.In Line [181], the authors mentioned "mechanical analysis". I didn't find it in the following content. Does it mean "mechanism analysis"?

5. In Section 3.3, the authors explored the design space of gold nanowire arrays and found the best structural configuration. My question here is why the authors utilizing 1.33-1.34 as the criteria to determine the best structure.

6. In Line [217], the authors separated the detection range into 3 ranges. It will be helpful if the authors could add some explanation about how the 3 ranges were determined. And for Figure 7, for each range, the wavelength vs RI curve has different fitting curves. Is there any theoretical explanation about this difference? The non-linear and various wavelength vs RI curve will bring challenge in real sensor applications because each sensor maybe calibrated independently before shipping.

7. In Table 1, the authors summarize the sensor performance in the detection range. I am curious if there is any fundamental limitation for the sensor to be utilized in RI>1.37 analytes. If not, that will extend the application for this sensor to many biomedical applications.

8.  There is no discussion part in this manuscript. I would extremely recommend the authors to include discussion such as what is the target potential applications of this sensor, the fabrication process of gold nanowire arrays and the advantage of proposed sensor configuration vs other RI detection methods like ring-resonator in a broader picture.  These discussions could help the readers to understand the significance of this work and guide the real sensor fabrication.

Author Response

Response to Reviewer 4 Comments

Manuscript ID: 1900449

Title: Fiber optic sensor with gold nanowire groups array for broad range and low refractive index detection

Dear Editors and Reviewers:

Thank you for your letter and for the reviewers’ comments concerning our manuscript entitled “Fiber optic sensor with gold nanowire groups array for broad range and low refractive index detection” (ID: 1900449).

Those comments are all valuable and very helpful for revising and improving our paper, as well as the important guiding significance to our researches. We have studied comments carefully and have made correction which we hope meet with approval. Revised portion are marked in red in the paper. The main corrections in the paper and the responds to the reviewer’s comments are as flowing. We appreciate for Editors/Reviewers’ warm work earnestly, and hope that the correction will meet with approval.

Once again, thank you very much for your comments and suggestions.

Response to Reviewer 4 Comments

Comments: In this manuscript, the authors studied a gold nanowire-based fiber optic sensor utilizing both PSPR and LSPR. From the simulation results, the proposed fiber optic sensor configuration helps improve the sensitivity of refractive index in the analytes. Overall, this work is thorough and interesting. It could potentially inspire the development of SPR based fiber optic sensors. However, I have the following comments and suggestions that hope authors can help address.

Point 1: In the introduction, it would be helpful if the authors could briefly specify what is the typical range of "low refractive index" and what is the typical definition of "broad range detection".

Response 1: We thank the reviewers for their valuable comments. It is true that "low refraction" and "wide range" are not explained in the introduction. The authors reviewed the relevant information and combined it with the model to make the appropriate explanation [1,2]. Generally speaking, the refractive index of water is 1.33, below 1.33 is called low refractive index. The detection range refers to the range of the sensor under the effective detection, the sensor proposed in this paper is to detect the refractive index of the analyte, wide range indicates that the detection of a larger range of refractive index, this does not have a hard and fast rules, with the analysis of the relevant literature on wide range detection, found that the general refractive index range of 0.10 or more is called wide refractive index, the detection range of this paper is 0.29, is a wide range of sensor. The authors have added the corresponding content in the article [2,3].

In lines 41-44, additional content “Generally speaking, pure water has a refractive index of 1.33, and analytes with refractive index lower than this value are called low-refractive analytes [1]. Wide range detection means that the effective detection range of the sensor is large, generally referring to the refractive index range greater than 0.10 [2,3]. ”

  1. Lee, K.J.; Liu, X.; Vuillemin, N.; Lwin, R.; Leon-Saval, S.G.; Argyros, A.; Kuhlmey, B.T. Refractive index sensor based on a polymer fiber directional coupler for low index sensing. Optics Express 2014, 22, 17497-17507, doi:10.1364/oe.22.017497.
  2. Li, X.; Warren-Smith, S.C.; Ebendorff-Heidepriem, H.; Zhang, Y.-n.; Nguyen, L.V. Optical Fiber Refractive Index Sensor With Low Detection Limit and Large Dynamic Range Using a Hybrid Fiber Interferometer. Journal of Lightwave Technology 2019, 37, 2954-2962, doi:10.1109/jlt.2019.2908023.
  3. Tan, X.-J.; Zhu, X.-S.; Shi, Y.-W. Hollow fiber sensor based on metal-cladding waveguide with extended detection range. Optics Express 2017, 25, 16996-17003, doi:10.1364/oe.25.016996.

Point 2: In Line [34], the authors claims "Compared to conventional detection, detection in the low refractive index (RI) range is often neglected by researchers, not only because there are fewer analytes to detect in this range, but also because it is more difficult to detect the low RI range, let alone the broad range detection of low refractive index [1,2]." This claim is self-contradictory with the content from Line [45] to Line [69] where the authors mentioned reference [11-21] numerous works has been studied in this field in the recent years. Also, if there are "fewer analytes" in this range, the significance of this work will be undermined. I suggest the authors to rephrase the wording here.

Response 2: We thank the reviewers for their valuable comments. In line 34, the authors want to convey that the detection technique of low refractive index with wide range is difficult and researchers need to work hard to overcome this difficulty. In lines 45 to 69, the authors want to express that the SPR-PCF sensor has been admired by researchers for its unique detection characteristics in recent years, but the research involved is not a sensor with both low refractive index and wide range. The authors want to build on the advantages of the SPR-PCF sensor to also achieve low refractive index and wide range detection. In the introductory section ambiguity arises due to the authors' logic and language failing to express it clearly. The authors have revised the relevant description in the introduction section to ensure that the article is logical and clearly expressed.

In lines 33-34, “Compared to conventional detection, detection in the low refractive index (RI) range is often neglected by researchers, not only because there are fewer analytes to detect in this range, but also because it is more difficult to detect the low RI range, let alone the broad range detection of low refractive index [1,2].” modified to “ The detection of low refractive index (RI) ranges is more difficult than conventional detection, and the detected range of lower RI is also smaller [1,2].”

Point 3: In Section 2.2 Principle Analysis, the authors provided an intuitive analysis of why PSPR+LSPR sensor has the potential for better sensing performance. Is it possible to include theoretical study to be more specific and help cross-validate with the simulation study in the following section?

Response 3: We thank the reviewers for their valuable comments. In Section 2.2, the authors illustrate the inclusion of PSPR and LSPR in the gold nanowire group array, which can enhance the detection capability of the sensor; the idea is the authors' intuitive analysis and lacks relevant theoretical support. The authors reviewed relevant information and combined the relevant theoretical analysis with the proposed sensor model to provide additional explanations for the relevant content of this section.

In lines 152-157, additional content “Our study of PSPR+LSPR sensing properties has been described in other related literature, where continuous and contacting Au nanowires can be used to generate PSPR and the upper and lower ends of each Au nanowire can generate LSPR, the combination of which achieves an effective enhancement of SPR [1,2]. In the following, we also analyze the relevant properties from simulations.”

  1. Ryu, J.-H.; Lee, H.Y.; Lee, J.-Y.; Kim, H.-S.; Kim, S.-H.; Ahn, H.S.; Ha, D.H.; Yi, S.N. Enhancing SERS Intensity by Coupling PSPR and LSPR in a Crater Structure with Ag Nanowires. Applied Sciences-Basel 2021, 11, doi:10.3390/app112411855.
  2. Bousiakou, L.G.; Gebavi, H.; Mikac, L.; Karapetis, S.; Ivanda, M. Surface Enhanced Raman Spectroscopy for Molecular Identification- a Review on Surface Plasmon Resonance (SPR) and Localised Surface Plasmon Resonance (LSPR) in Optical Nanobiosensing. Croatica Chemica Acta 2019, 92, 479-494, doi:10.5562/cca3558.

Point 4: In Line [181], the authors mentioned "mechanical analysis". I didn't find it in the following content. Does it mean "mechanism analysis"?

Response 4: Thanks to the reviewers for pointing out the errors. Due to the author's oversight, 'mechanical analysis' was written as 'mechanical analysis'. The authors have corrected the error.

In lines 214-215, “We illustrate the advantages of Au nanowire group array through mechanical and simulation analyses.” modified to “We illustrate the advantages of Au nanowire group array through mechanism and simulation analyses.”

Point 5: In Section 3.3, the authors explored the design space of gold nanowire arrays and found the best structural configuration. My question here is why the authors utilizing 1.33-1.34 as the criteria to determine the best structure.

Response 5: We thank the reviewers for their valuable comments. In section 3.3, we mainly divided the effect of the diameter of gold nanowires on the sensitivity. We chose the sensitivity of 1.33-1.34 for the analyte for two main reasons. First, it is beneficial to observe the relationship between the diameter of gold nanowires and the sensitivity more intuitively. Due to the limitation of simulation accuracy, if the sensitivity changes are small, the relationship between the two will not be observed very accurately, and it is known from the simulation data that the lower the refractive index of the analyte, the smaller the sensitivity, so the refractive index of the analyte cannot be too low. In addition it can not be too high, because too high refractive index may not be universal in the entire detection range. Second, it facilitates the validation of the sensor after manufacture. If the sensor is manufactured, we need to compare the experimental data and simulation data for analysis, so in the selection of analytes to choose easy to obtain and relatively stable substances. Pure water is an ideal analyte with a refractive index of 1.33. Therefore, for easier experimental validation, the refractive index range of 1.33-1.34 is used in the simulation. The authors add the reasons for the choice of refractive index in Section 3.3 of the paper.

In lines 249-251, additional content “To ensure a clearer observation of the relationship between Au nanowire diameter and sensitivity, and to ensure a more suitable analyte for the subsequent experimental phase, we chose an analyte with a refractive index of 1.33 to 1.34.”

Point 6: In Line [217], the authors separated the detection range into 3 ranges. It will be helpful if the authors could add some explanation about how the 3 ranges were determined. And for Figure 7, for each range, the wavelength vs RI curve has different fitting curves. Is there any theoretical explanation about this difference? The non-linear and various wavelength vs RI curve will bring challenge in real sensor applications because each sensor maybe calibrated independently before shipping.

Response 6: Thank you for your valuable comments. Our proposed sensor implements a wide range of detection from refractive index 1.08 to 1.37, which would affect the accuracy of the data if all the data were put together for analysis, so we divided the data into three parts. The three parts are divided on the basis of an equal distribution with a slight adjustment according to the fitted curve. This is because a higher fit will more accurately reflect the detection of the sensors. As can be seen from Table I, the sensitivity of the sensor does not increase linearly and regularly, while the fitted curves in all three ranges ensure a higher fit, hence the difference between them. Admittedly, a nonlinear fit curve does increase the manufacturing difficulty. But we are only at the simulation stage, where a better-fitting fit curve can more accurately describe the sensor detection performance, and that is what we are more concerned with at that stage. In the subsequent experimental stage or even the industrial manufacturing stage, many factors will cause its data to be different from the simulation data, so in this stage many parameters will be modified according to the actual requirements, so as to achieve the industrial production of the sensor.

Point 7: In Table 1, the authors summarize the sensor performance in the detection range. I am curious if there is any fundamental limitation for the sensor to be utilized in RI>1.37 analytes. If not, that will extend the application for this sensor to many biomedical applications.

Response 7: We thank the reviewers for their suggestions. The main theme of the research in this paper is the wide range detection of low refractive indices, with a refractive index of 1.33 or less locating low refractive indices. Since the detection capability of the sensor remains stable in the range of 1.33 to 1.37, we have selected 1.08-1.37 as the effective detection range. This does not mean that the refractive index above 1.37 is undetectable; during the simulation, the data above the refractive index of 1.37 appears to fluctuate slightly. The main objective of our research is to propose a fiber optic sensor with PSPR and LSPR sensing characteristics. The detection range can be controlled according to the structure of the fiber, the material, and the various parameters of the metal nanowire array, so it is perfectly possible to design a suitable detection range for biological detection, and we will make subsequent research in this direction.

Point 8: There is no discussion part in this manuscript. I would extremely recommend the authors to include discussion such as what is the target potential applications of this sensor, the fabrication process of gold nanowire arrays and the advantage of proposed sensor configuration vs other RI detection methods like ring-resonator in a broader picture. These discussions could help the readers to understand the significance of this work and guide the real sensor fabrication.

Response 8: We thank the reviewers for their comments. The article mainly uses comparative analysis to analyze the detection characteristics of our proposed sensor, and there is less discussion in other related directions. Therefore, we have added manufacturing and application discussion and analysis in the article, taking into account the comments given by the reviewers.

In lines 286-295, additional content “The fabrication of the sensor proposed in this paper is not complicated. Firstly, we selected pure silicon rods of different diameters and made some of them into hollow silica capillaries, then stacked them together in a circle and stretched them at high temperature. Next, a detection plane was polished on one side of the stretched fiber, then a TiO2 layer was coated on that plane, and finally, an array of Au nanowires array fabricated by combining the DC magnetron sputtering, dip coating and self-assembly evaporation techniques were transferred to the sensing plane [1,2]. Our proposed Au nanowire group array structure is well applied in the field of bioinformatics detection. Besides, the effect of the structure on SPR can also be applied to solar cells [3], photoelectric conversion [4], photoelectric switches [5] and other fields, and its has good application value. ”

  1. Wieduwilt, T.; Kirsch, K.; Dellith, J.; Willsch, R.; Bartelt, H. Optical Fiber Micro-Taper with Circular Symmetric Gold Coating for Sensor Applications Based on Surface Plasmon Resonance. Plasmonics 2013, 8, 545-554, doi:10.1007/s11468-012-9432-7.
  2. Jing, G.; Bodiguel, H.; Doumenc, F.; Sultan, E.; Guerrier, B. Drying of Colloidal Suspensions and Polymer Solutions near the Contact Line: Deposit Thickness at Low Capillary Number. Langmuir 2010, 26, 2288-2293, doi:10.1021/la9027223.
  3. Zhang, J.-C.; Mao, Y.; Wang, W.; Guan, Y.-R.; Bao, Y.; Niu, L. In Situ Investigation on Electrochemical Polymerization and Properties of Polyaniline Thin Films by Electrochemical Surface Plasmon Resonance. Chinese Journal of Analytical Chemistry 2015, 43, 350-355.
  4. Tan, D.; Wang, Y.; Gan, Y. Facile Visible-Light-Assisted Synthesis, Optical, and Electrochemical Properties of Pd Nanoparticles with Single crystalline and Multiple-twinned Structures. Rare Metal Materials and Engineering 2017, 46, 2065-2069.
  5. Baba, A.; Tada, K.; Janmanee, R.; Sriwichai, S.; Shinbo, K.; Kato, K.; Kaneko, F.; Phanichphant, S. Controlling Surface Plasmon Optical Transmission with an Electrochemical Switch Using Conducting Polymer Thin Films. Advanced Functional Materials 2012, 22, 4383-4388, doi:10.1002/adfm.201200373.

Reviewer 5 Report

The authors investigated the sensor for refractive index based on plasmonic fiber. It concludes that PSPR+LSPR enhance the sensitivity. The result looks good compared with similar design. Here are my comments.

On page 7, authors claimed that detection range is enhanced, which is supported by any analysis. In fact, the detection range can be limited by several factors. E.g. the range of wavelength scan, resolution bandwidth, and the flatness of the resonance peak. Without detailed data, it is hard to conclude the Au nanowire groups array can enhance the detection range.

On page 6, it should be Figures 5(a)-(d) instead of Figure 4(a)-(d).

Author Response

Response to Reviewer 5 Comments

Manuscript ID: 1900449

Title: Fiber optic sensor with gold nanowire groups array for broad range and low refractive index detection

Dear Editors and Reviewers:

Thank you for your letter and for the reviewers’ comments concerning our manuscript entitled “Fiber optic sensor with gold nanowire groups array for broad range and low refractive index detection” (ID: 1900449).

Those comments are all valuable and very helpful for revising and improving our paper, as well as the important guiding significance to our researches. We have studied comments carefully and have made correction which we hope meet with approval. Revised portion are marked in red in the paper. The main corrections in the paper and the responds to the reviewer’s comments are as flowing. We appreciate for Editors/Reviewers’ warm work earnestly, and hope that the correction will meet with approval.

Once again, thank you very much for your comments and suggestions.

Response to Reviewer 5 Comments

Comments: The authors investigated the sensor for refractive index based on plasmonic fiber. It concludes that PSPR+LSPR enhance the sensitivity. The result looks good compared with similar design. Here are my comments.

Point 1: On page 7, authors claimed that detection range is enhanced, which is supported by any analysis. In fact, the detection range can be limited by several factors. E.g. the range of wavelength scan, resolution bandwidth, and the flatness of the resonance peak. Without detailed data, it is hard to conclude the Au nanowire groups array can enhance the detection range.

Response 1: Thanks to the reviewer's comments, the authors analyzed the metal nanowire group array structure with less description and data about the increased detection range and lack of relevant data support, in addition to the fact that there are more factors affecting the detection range and we have to prove that it is the proposed structure that increases the detection range and not other factors. The authors have done simulations for the redoing of the detection range and added the results and the related discussion to the paper.

Figure 5. Analysis of the detection performance of different types of sensing areas. Sensitivity analysis (a) Au thin layer; (b) Au nanowires; (c) Au nanowire groups array with  spacing; (d) Au nanowire group array with  and  spacing. Detection range analysis (e) Au thin layer, continuous Au nanowires and Au nanowire groups array.

In lines 230-240, additional content “In addition to the improvement in sensitivity, the Au nanowire groups array can also achieve an improvement in detection range. As shown in Figure 5(e), when the refractive index is 1.15, the loss spectra of Au thin layer, continuous Au nanowires and Au nanowire groups array are analyzed under the premise of ensuring the same substitution parameters. When the refractive index is low, the detection ability of Au thin layer is limited, and its loss curve is monotonically increasing, and the change is not very obvious to achieve the detection in this refractive index. Continuous Au nanowire structure in this refractive index there is a significant fluctuation, but did not form a clear wave peak, so also can not be well achieved detection. The Au nanowire group array structure exhibits a clearer wave peak at this refractive index, and although the data are not very good, they are effective peaks at low refractive indices. Therefore, the Au nanowire group array designed by the authors can appropriately increase the detection range of the sensor.”

Point 2: On page 6, it should be Figures 5(a)-(d) instead of Figure 4(a)-(d).

Response 2: Thanks to the reviewer for pointing out the error. Due to an oversight of the authors, “Figures 5(a)-(d)” was written as “Figures 4(a)-(d)”. The authors have corrected the error.

In lines 215-216, “Figures 4(a)-(d) show the numerical analysis of the induction zone with different structures.” modified to “Figures 5(a)-(d) show the numerical analysis of the induction zone with different structures.”

Round 2

Reviewer 2 Report

Although the author addresses all the concerns, it still has some minor issues listed below:

Query 1. The state of the art is improved but not explained clearly as it fails to discuss the need for proposed work over other similar technology. The author mentioned that Pathak, A.K. et al. reported 90500 nm/RIU (which is extremely high) sensitivity but did not discuss the results of Meshginqalam et al. and Zhan, Y. et al. work. Authors are advised to discuss the drawbacks/limitations of these sensors that can be resolved by the author's structure.

Query 2. In table 2, I still can see a comparison of sensing response with thin film-based sensors. Authors are advised to remove the sensing response of thin film-based sensors and compare it with nanowire-based sensors.

Query 3. In the previous version, the kink appears not at the phase matching point but far away from it, now with the current simulation, the kink disappears in the revised version (Fig. 3) which means there was some parameter or meshing issue with the simulation. My concern is regarding the new data, does this new simulation affect the previous optimized value? if not then it's ok, and if yes, please mention it in the revised manuscript.

Author Response

Response to Reviewer 2 Comments

Manuscript ID: 1900449

Title: Fiber optic sensor with gold nanowire groups array for broad range and low refractive index detection

Dear Editors and Reviewers:

Thank you for your letter again and for the reviewers’ comments concerning our manuscript entitled “Fiber optic sensor with gold nanowire groups array for broad range and low refractive index detection” (ID: 1900449).

Those comments are all valuable and very helpful for revising and improving our paper, as well as the important guiding significance to our researches. We have studied comments carefully and have made correction which we hope meet with approval. We still keep the first revision and mark it in red, and this revision is marked in blue. The main corrections in the paper and the responds to the reviewer’s comments are as flowing. We appreciate for Editors/Reviewers’ warm work earnestly, and hope that the correction will meet with approval.

Once again, thank you very much for your comments and suggestions.

Response to Reviewer 2 Comments

Comments: Although the author addresses all the concerns, it still has some minor issues listed below:

Point 1: The state of the art is improved but not explained clearly as it fails to discuss the need for proposed work over other similar technology. The author mentioned that Pathak, A.K. et al. reported 90500 nm/RIU (which is extremely high) sensitivity but did not discuss the results of Meshginqalam et al. and Zhan, Y. et al. work. Authors are advised to discuss the drawbacks/limitations of these sensors that can be resolved by the author's structure.

Response 1: Thank you for your valuable comments again. The author did supplement the technical analysis with only the Pathak, A.K. et al. work, as this article performed better and the other two articles are referenced in other parts of the article. The authors supplemented the technical analysis with relevant analyses from Meshginqalam et al. and Zhan, Y. et al. work based on the reviewers' comments.

In lines 83-84, additional content “The detection sensitivity for MCF-7 was 53571 nm/RIU.”

In lines 86-87, additional content “It has a detection range of 1.33-1.40 and a maximum sensitivity of 12314 nm/RIU.”

In lines 87-90, “Sensor has potential applications in biochemical detection.” modified to “These metal nanowire structures offer a greater improvement in detection performance than conventional thin-layer sensors. However, they also have some drawbacks, such as how to fix the nanowires smoothly and how to enhance the detection range of this type of sensor.”

In line 90, additional content “The sensor is D-type and the detection plane is easier to fix gold nanowires.”

Point 2: In table 2, I still can see a comparison of sensing response with thin film-based sensors. Authors are advised to remove the sensing response of thin film-based sensors and compare it with nanowire-based sensors.

Response 2: Thank you for your valuable comments again. It is true that some thin layer structures exist in Table 2 for comparative analysis. The authors modified some of the comparison information in Table 2 by using more metal nanowires for the sensors.

Table 2. Performance analysis of the same type of sensors.

Design

ns range

Wavelength range (nm)

Max WS

(nm/RIU)

Ave.WS

(nm/RIU)

Au nanowires [1]

1.33-1.36

599-768

9000

5500

Au nanowires Low RI [2]

1.27-1.36

640-960

9100

2350

Ag nanowire, Broad Range [3]

1.33-1.43

530-920

10300

4300

Ag nanowire [4]

1.33-1.38

1097-1570

9314

4730

Low RI, Broad Range,

PSPR, LSPR

1.08-1.16

1210-1300

2000

1125

1.17-1.27

1320-1510

4000

1900

1.28-1.37

1540-2140

13000

6667

  1. Jiao, S.; Gu, S.; Yang, H.; Fang, H. Research on dual-core photonic crystal fiber based on local surface plasmon resonance sensor with silver nanowires. Journal of Nanophotonics 2018, 12, doi:10.1117/1.Jnp.12.046015.
  2. Liu, C.; Yang, L.; Liu, Q.; Wang, F.; Sun, Z.; Sun, T.; Mu, H.; Chu, P.K. Analysis of a Surface Plasmon Resonance Probe Based on Photonic Crystal Fibers for Low Refractive Index Detection. Plasmonics 2018, 13, 779-784, doi:10.1007/s11468-017-0572-7.
  3. Zhao, L.; Han, H.; Lian, Y.; Luan, N.; Liu, J. Theoretical analysis of all-solid D-type photonic crystal fiber based plasmonic sensor for refractive index and temperature sensing. Optical Fiber Technology 2019, 50, 165-171, doi:10.1016/j.yofte.2019.03.013.
  4. Pathak, A.K.; Singh, V.K. SPR Based Optical Fiber Refractive Index Sensor Using Silver Nanowire Assisted CSMFC. Ieee Photonics Technology Letters 2020, 32, 465-468, doi:10.1109/lpt.2020.2980470.

Point 3: In the previous version, the kink appears not at the phase matching point but far away from it, now with the current simulation, the kink disappears in the revised version (Fig. 3) which means there was some parameter or meshing issue with the simulation. My concern is regarding the new data, does this new simulation affect the previous optimized value? if not then it's ok, and if yes, please mention it in the revised manuscript.

Response 3: Thank you for your comments again. In the original manuscript, the kink appears not at the phase matching point also due to the lack of precision in simulation and plotting. In the first revision of the manuscript, the authors have improved the accuracy, so the kink is reduced or even disappeared. This change did not affect the related data.
